# On the relationship between mesoscale cellular convection and meteorological forcing: Comparing the Southern Ocean against the North Pacific

Francisco Lang[1,2], Steven T. Siems[2,3], Yi Huang[4,5], Tahereh Alinejadtabrizi[2,3,5], and Luis Ackermann[6]

[1]Department of Geophysics, Universidad de Concepción, Concepción, Chile.
[2]School of Earth, Atmosphere and Environment, Monash University, Melbourne, Victoria, Australia.
[3]Australian Research Council Securing Antarctica's Environmental Future (SAEF), Melbourne, Victoria, Australia.
[4]School of Earth Sciences, The University of Melbourne, Melbourne, Victoria, Australia.
[5]Australian Research Council Centre of Excellence for Climate Extremes (CLEX), Melbourne, Victoria, Australia.
[6]Australian Bureau of Meteorology, Melbourne, Victoria, Australia

**Correspondence:** Francisco Lang (flang@udec.cl)

**Abstract.** Marine atmospheric boundary layer (MABL) clouds cover vast areas over the ocean and have important radiative effects on the Earth's climate system. These radiative effects are known to be sensitive to the local organization, or structure, of the mesoscale cellular convection (MCC). A convolution neural network model is used to identify the two idealized classes of MCC clouds, namely open and closed, over the Southern Ocean (SO) and Northwest Pacific (NP) from high-frequency geostationary Himawari-8 satellite observations. The results of the climatology show that MCC clouds are evenly distributed over the mid-latitude storm tracks for both hemispheres, with peaks poleward of the 40° latitude. Open MCC clouds are more prevalent than closed MCC in both regions. An examination of presumed meteorological forcing associated with open and closed MCC clouds is conducted to illustrate the influence of large-scale meteorological conditions. We establish the importance of the Kuroshio western boundary current in the spatial coverage of open and closed MCC across the NP, presumably through the supply of strong heat and moisture fluxes during marine cold air outbreaks events. In regions where static stability is higher we observe a more frequent occurrence of closed MCCs. This behavior contrasts markedly with that of open MCCs, whose formation and persistence are significantly influenced by the difference in temperature between the air and the sea surface. The occurrence frequency of closed MCC over the SO exhibits a significant diurnal cycle, while the diurnal cycle of closed MCC over the NP is less noticeable.

## 1 Introduction

Marine atmospheric boundary layer (MABL) clouds have a significant contribution to the energy budget over the oceans (Trenberth and Fasullo, 2010), covering nearly 25% of the marine surface (Wood, 2012). Small changes in the spatial distribution or

physical properties of these clouds are able to produce important radiative effects in the high and mid-latitudes (Bodas-Salcedo et al., 2016). Given the remote location of these clouds, satellite observations have been indispensable for advancing our understanding of them. The first satellite observations in the early 1960s (e.g., Agee and Dowell, 1974; Atkinson and Zhang, 1996) revealed that these shallow clouds commonly take the form of open or closed mesoscale cellular convection (MCC), which are defined by distinct patterns of organization. These clouds are particularly important to the climate, as they help regulate both the shortwave and long-wave fluxes as well as sensible and latent heat fluxes into and out of the ocean. For example, closed MCC clouds over the Southern Ocean (SO) have been found to have on average a higher albedo than open MCC clouds for the same cloud fraction, which can drive changes in the cloud radiative effect up to 39 W m$^{-2}$, depending on season and cloud phase (Danker et al., 2022; McCoy et al., 2017, 2023). Zelinka et al. (2020) showed that the climate simulations are most sensitive to low-level clouds across these latitudes. Wood and Hartmann (2006) first developed a cloud classification algorithm to classify low-level clouds scenes from satellite observations over the eastern subtropical Pacific Ocean into four categories based on the level of cellularity and mesoscale organization: open, closed, cellular but disorganized MCCs and no MCC present. Their method was based on the training of a two-layer neural network on probability distribution functions and 2-D power spectra of liquid water path; their analysis, however, was limited to only 2 months of data over a region limited to warm clouds. More recent investigations of MCC cloud classification have more comprehensively examined the Earth's oceans using machine learning (e.g., Rampal and Davies, 2020; Watson-Parris et al., 2021; Yuan et al., 2020), confirming that these various MCC cloud formations dominate in the mid-latitude storm tracks (Agee et al., 1973; Lang et al., 2022; McCoy et al., 2023; Muhlbauer et al., 2014). Lang et al. (2022) built a climatology of open and closed MCC over the Australian sector of the SO using geostationary satellite data from Himawari-8 and a convolutional neural network. An advantage of this methodology was that spatial inhomogeneities, from sea surface temperature (SST) gradients to orography, became apparent.

Over the mid-latitudes, open MCC clouds are found to be commonly associated with the cold air sector of extra-tropical cyclones and the ensuing marine cold air outbreaks (MCAO) (McCoy et al., 2017). Across the subtropics, conversely, closed MCC clouds are more prevalent in the stratocumulus decks west of continents (Klein and Hartmann, 1993). Previous studies have identified multiple large-scale meteorological and thermodynamic factors that can influence MCC cloud development. For instance, open MCC clouds are particularly influenced by surface forcing, while closed MCC clouds are more responsive to longwave cloud top cooling (McCoy et al., 2017; Wood, 2012). However, this distinction becomes less apparent in subtropical regions, where the absence of significant meteorological differences between open and closed MCC clouds suggests a predominant role of the precipitation mechanism. This finding indicates a potential zonal variation in the importance of these influencing factors. Closed MCCs commonly transition into open MCCs and disorganized MCCs in the subtropics (e.g., Eastman et al., 2022; Yamaguchi et al., 2017). One mechanism driving this transition is advection over warmer water, where drizzle leads to the breakup of closed MCCs into open MCC clouds (Kazil et al., 2011; Yamaguchi et al., 2017). In the high latitudes and mid-latitudes, similar transitions are associated with even stronger surface forcing during MCAOs (e.g., Abel et al., 2017; Atkinson and Zhang, 1996; Fletcher et al., 2016a; McCoy et al., 2017). Both categories of MCC clouds exhibit a distinct geographical distribution and a pronounced seasonal variation (e.g., Lang et al., 2022; Muhlbauer et al., 2014).McCoy et al. (2017) found that the seasonal cycle of open MCC clouds has a strong connection with the MCAOs in the mid-latitudes,

while the seasonal cycle of closed MCC clouds is influenced by surface forcing in the mid-latitudes and lower tropospheric stability in the tropics and subtropics. Over mid-latitudes, MCAOs influence not only the development of open MCC clouds but also the transition from closed to open MCC clouds. MCAO events are characterized by strong latent and sensible heat fluxes that drive the MABL convection and the distinct mesoscale organization of these clouds. Strong MCAOs are characterized by greater cloud fraction and optical thickness than weak ones, resulting in a greater shortwave cloud radiative effect (Fletcher et al., 2016b). Fletcher et al. (2016a) investigated MCAOs in both hemispheres and found that they are more vigorous and more frequent in the Northern Hemisphere (NH) than the Southern Hemisphere (SH). Tornow et al. (2021) investigated the influence of frozen hydrometeors on these transitions during a case of MCAO within the Aerosol Cloud meTeorology Interactions oVer the western ATlantic Experiment (ACTIVATE) campaign. Their research revealed that ice particles significantly expedite these transitions while also decreasing the amount of cloud liquid water.

Comparisons of the mid-latitude storm track between the two hemispheres have found a variety of differences beyond MCAOs. Huang et al. (2015) used reanalysis datasets and A-Train observations to show differences in the storm-track cloud properties. They found that the North Atlantic has a stronger seasonality in cloud properties than the SO. In summer, boundary layer cloud heights between the two regions are comparable, while the wintertime North Atlantic is dominated by higher boundary layer clouds than the SO. With CloudSat/CALIPSO observations,Muhlbauer et al. (2014) showed that the seasonal cycle of low-level cloud fraction peaks during boreal summer in the mid-latitude storm track regions of the North Atlantic and North Pacific while a seasonal cycle is almost absent in the SO. The lack of seasonal cycle in the low-level cloud fraction over the SO is consistent with the findings in (Huang et al., 2012a) based on sounding observations over Macquarie Island. MCC clouds systems have also shown difference between the NH and the SH environments. McCoy et al. (2017) used Moderate Resolution Imaging Spectroradiometer (MODIS) data and the European Center for Medium-Range Weather Forecasting (ECMWF) ERA-Interim data to demonstrate that surface fluxes and inversion strength are both important for these cloud types; however, over the mid-latitudes, surface forcing appears to be much more important to closed MCC in the NH than in the SH. This surface forcing undergoes a much stronger seasonal cycle over the NH reflecting the larger cycle in the temperature of terrestrial air masses in the NH. Another substantial difference between the NH and the SH is aerosol characteristics, which can alter cloud structure, microphysical, and radiative properties. For example, satellite retrievals of cloud-top phase indicate that supercooled liquid water is more prevalent over the SO than at equivalent latitudes in the NH (Choi et al., 2010; Hu et al., 2010; Morrison et al., 2011). Using MODIS observations, Huang et al. (2016) also found that the average cloud droplet number concentration is substantially greater over the North Pacific than the mid-latitude SO.

In this work, we extend the climatology of Lang et al. (2022)to the Northwest Pacific (NP), which is also covered by Himawari-8 observations. The objective of this study is to investigate the differences between open and closed MCC cloud distributions and their relationship with large-scale meteorological and thermodynamic forcings over the NP and the SO. First, we consider the Estimated Inversion Strength (EIS) calculated from the ECMWF ERA5 reanalysis. Naud et al. (2016) demonstrated that EIS is well correlated to cloud cover in dynamically active sectors after cold front passages. Next, we examine the relationship with respect to the MCAO index, $M$ (Fletcher et al., 2016a), defined as the difference between the ocean potential skin temperature and the 800 hPa potential temperature. McCoy et al. (2017) found that the $M$ index is a good

predictor of MCC clouds organization in the extra-tropical oceans. In addition, the spatial correlation between both categories of MCC and the meteorological indices $M$ and EIS, as well as near-surface wind speed and SST, is also analyzed. Again, taking advantage of the geostationary platform our analysis focusses on differences between the NP and the SO.

## 2   Data and methodology

### 2.1   Data source and domain

The classification dataset used is derived from the Advanced Himawari Imager (AHI) on board the Himawari-8 geostationary meteorological satellite (Bessho et al., 2016). Himawari-8 was launched in July 2015 by the Japanese Meteorological Agency. Himawari-8 scenes and cloud products are available on the Japan Aerospace Exploration Agency (JAXA) P-Tree System. Himawari-8 provides a temporal resolution of 10 min and spatial resolutions ranging from 1 to 5 km. Himawari-8 products include visible, infrared, and water vapor imagery, as well as daytime cloud products such as cloud types, cloud-top properties, cloud effective radius and cloud optical thickness. Hourly brightness temperature from infrared channel 11 (8.6 $\mu$m) at 5 km resolution in an orthogonal gridded projection was used for the model training and subsequent MCC climatology classification.

For this study, the analysis is based on 3 years (2016-2018) of Himawari-8 scenes from the SH region (20-60° S, 80°E- 160° W) which includes the SO and portions of the Pacific and Indian oceans, as well as scenes from the NP region (20-60° N, 80°E- 160° W). Centered at 140.7° E the satellite covers the Asia-Oceania region. Both domains encompass the area of the storm tracks in the mid-latitude that are directly associated with MCAOs. Reanalysis data are employed in this study to examine the connections between large-scale meteorological and thermodynamic forcings and MCC clouds. We used the ECMWF ERA5 reanalysis (Hersbach et al., 2020). Hourly data gridded to $0.75 \times 0.75°$ grid boxes are used over the two regions between 2016 and 2018.

### 2.2   Cloud type classification

The Lang et al. (2022) classification scheme of MABL clouds is based on a hybrid convolutional neural network (CNN) model defined for the two primary classes of MCC clouds: open and closed. By default, the algorithm includes a third category called "other" that it is used for all other coverage including mid- and high-level clouds, as well as other types of low-level cloud types such as stratus, disorganized MCC, and even clear skies. Hourly brightness temperature at 5 km resolution from infrared channel 11 (8.6 $\mu$m) is used as the main input to train the CNN. To build up the labelled training data set, additional Himawari-8 channels and products were used as contextual information and filtering. To maintain labeling consistency for open and closed MCCs, a stringent criterion is applied in Lang et al. (2022), ensuring that the structure of the MCC clouds adheres to specific characteristics. Open MCCs must exhibit a well-formed open cell cloud structure, forming interconnected open rings, while closed MCCs must display a closed cell cloud structure, appearing more "bubbly" in nature (Watson-Parris et al., 2021). A full description of the machine learning training and performance evaluation can be found in Lang et al. (2022). Examples of training samples for the three categories can be found in Fig. 2 of Lang et al. (2022). The classification algorithm developed in

120 Lang et al. (2022) found an average precision of about 89 % across all categories. Open MCC had the lowest accuracy, most likely because it had the smallest training sample size. The largest source of uncertainty reported by Lang et al. (2022) was the difficulty to separate open from disorganized MCC, a challenge similarly discussed in Yuan et al. (2020). An example of a classified image from Himawari-8 during the winter over the NP mid-latitude is shown in Fig. 1. The scene shows a frontal band moving eastward followed by groups of open and closed MCC clouds in a post-frontal environment. Closed MCCs are 125 primarily observed to the northwest of open MCCs, further away from a cold front.

In relation to the category "other", available Himawari-8 cloud products (daytime) are utilized to distinguish low-level clouds within this category. A filter based on cloud-top height is applied to identify clouds below 3.5 km within the category "other".

## 2.3 Large-scale meteorological indices

Two meteorological indices are employed in our study, the EIS (Wood and Bretherton, 2006) and the MCAO index ($M$, Fletcher 130 et al., 2016a). These indices represent different features of large-scale dynamic and thermodynamic influences on open and closed MCC cloud development. EIS is a measure of the strength of the boundary layer inversion, an indicator of the static stability of the lower atmosphere. It is defined as follows:

$$EIS = LTS - \Gamma_m^{850}\left(z_{700} - LCL\right), \tag{1}$$

where $LTS$ is the lower tropospheric stability defined in (Klein and Hartmann, 1993), which is the difference in potential 135 temperature between 700 hPa and the surface ($LTS = \theta_{700} - \theta_{surf}$). The variable $\Gamma_m^{850}$ is the moist-adiabatic potential temperature gradient at 850 hPa; $z_{700}$ is the altitude of the 700-hPa level, and $LCL$ is the lifting condensation level. Large EIS is associated with strong and low-lying inversions, which may favor trapping moisture within the MABL more efficiently and promoting greater cloud cover (Lang et al., 2018; Kawai et al., 2017). Figure 2 shows the frequencies of occurrence of the EIS estimated from the ERA5 reanalysis products between 2016 and 2018, for the region between 60° N-60° S and 80° E-160° 140 W, which corresponds to the area of the Himawari-8 full disk. For both hemispheres, the highest EIS values are primarily observed over the subtropics and part of the mid-latitude regions, with a local maximum over the southeastern Indian Ocean in the SH and over the Sea of Okhotsk in the NH. Both areas have been associated with high percentages of low-cloud fraction (Muhlbauer et al., 2014).

The MCAO index $M$ is defined as the difference between the surface skin potential temperature and the 850-hPa potential 145 temperature:

$$M = \theta_{SKN} - \theta_{850}, \tag{2}$$

Fletcher et al. (2016a, b) based the index $M$ on the surface skin temperature and not the SST, to exclude areas of high sea ice cover. Fletcher et al. (2016a) used the potential temperature at 800 hPa, while Papritz et al. (2015) used 850-hPa potential temperature over the South Pacific. We tested both potential temperature levels and found that using the 800-hPa

level produced far fewer MCAOs over the SO than using the 850-hPa level, which makes the 850-hPa potential temperature level a more appropriate value to compare the two regions. Following Fletcher et al. (2016a) we define MCAOs as contiguous oceanic regions where $M > 0$ K. Figure 2b shows the frequency of occurrence of $M$. The relative frequency was defined by dividing the amount of time where $M > 0$ by the extent of the full record. While EIS and $M$ capture distinct characteristics, they are not entirely independent of each other.

For both hemispheres, MCAOs most often occur in areas with a strong SST gradient over the mid-latitude (Figure 2c), most notably the Kuroshio region, east of Japan, with a frequency of 50%. This is consistent with the results found in Fletcher et al. (2016b) for $M$ index using ERA-Interim data. The 3-yr mean SST composite is shown in Fig. 2c. In general,the NH SSTs are greater than the SH counterparts for the same latitude bands. Over the SO mid-latitudes, the oceanic polar front is a pronounced feature, associated with the strongest meridional SST gradients (Dong et al., 2006; Lang et al., 2022; Truong et al., 2020). The full-year SST gradients in the NP ocean, on the other hand, are weaker.

## 3   Results

Following (Lang et al., 2022), the period analyzed covers the hourly brightness temperature images from Himawari-8 for the three years between 2016 and 2018, for both the SO and NP regions. For this period, 25,494 images were processed and classified into each category (open MCC, closed MCC, and other) and $\sim 400$ were used as the training dataset (Lang et al., 2022). We note that the training data came exclusively from the SO. A visual inspection of several cases and time periods where open and closed MCC clouds are present over the NH storm track confirmed that the algorithm was producing robust results for both hemispheres.

### 3.1   MCC climatology: North Pacific versus Southern Ocean

The annual frequency of MCC cloud occurrence is defined as the number of times a specific cloud category is observed at a grid point, divided by the total number of observations (Fig. 3). Results for the SO have previously been discussed in Lang et al. (2022). Focusing on the NP here, we observe peak frequencies of both open and closed MCC poleward of $40°$ N latitude with the peak in closed MCC being slightly closer to Asia ($\sim 160°$ E) than for the open MCC ($\sim 180°$ E). Strong gradients in the frequency of open MCC are apparent to the north and south of this band of open MCC, with few present equatorward of $30°$ N and poleward of $55°$ N. A small local peak is also observed just to the east of the Tohoku Prefecture in Japan. Closed MCCs are primarily located at poleward of $40°$ N with small peaks in the Sea of Japan, the Sea of Okhotsk and the Bering Sea. For open MCCs, the frequency exceeds 20% in a band across much of the SO mid-latitudes compared to 13% across the mid-latitude in the NP. Overall, we observe the peak frequency of open MCC to be much greater over the SO ($\sim 25$ %) than over the NP ($\sim 16$ %). Conversely for closed MCC, we observe little difference in the peak frequency over the SO (11 %) and the NP ($\sim 13$ %), but this overlooks an important difference between the hemispheres. Over the SO, a closed MCC peak is located over the southeast Indian Ocean, off the coast of southwest Western Australia. This is a region dominated by sub-tropical subsidence (Atkinson and Zhang, 1996) rather than the SO storm track, and is also influenced by the colder SST

due to the Western Australian current. Looking at the higher latitude band (40 - 60°), we find more closed MCC over the NP than the SO. Closed MCC is evident near the top of the domain over the NP, but is relatively sparse at high latitudes of the SO, poleward of the ocean polar front.

We can look back at the spatial relationship between the annual frequency of open and closed MCC (Fig. 3) against the EIS, $M$ index, average SST (Fig. 2, top row), and wind speeds (Fig. S1 in Supplementary Material) from ERA5. Figure 4 presents correlation maps between monthly MCC cloud frequency and monthly averages of $M$ index, EIS, SST, and near-surface wind speed. Correlations were computed using the monthly mean MCC cloud frequencies in each grid box, and they are statistically significant at a 95% confidence level for a 36-point (three year) correlation. Correlations for individual grid point samples of each variable are displayed in Figure S2. Correlations were also calculated using the daily mean MCC cloud frequencies against daily averages of the four meteorological variables (Figure S3). These correlations show that, although they exhibit more noise compared to those on a monthly scale, the spatial patterns still align with the monthly correlations.

The results show that the largest coherent pattern of correlation with open MCC clouds is located over the mid-latitude storm tracks, roughly between latitudes 30 to 50°, for most of these variables. The correlations for open MCC clouds exhibit a strong relationship across the NP following the Kuroshio current, as well as across the SO. For both near-surface wind speed and $M$ index correlation coefficients, a robust pattern is evident across both study regions, encompassing a zonal band that spans the entire width of both domains. The correlations between open MCC and $M$, as well as between open MCC and wind speeds, exhibit notable similarity in both regions. In the case of correlations between SST and open MCCs, both regions show a clear separation between positive and negative correlations. Stronger correlations are observed at lower latitudes, whereas at higher latitudes the correlations show the opposite trend. It is interesting to note that the boundaries between positive and negative correlation coefficients seemingly correspond to the polar front zones, as demonstrated in Fig. S1. Conversely, we find little significant correlation for the closed MCC across the storm tracks, particularly the SO storm track, for any of the four variables. One exception, however, is the correlation between closed MCC frequency and EIS over the NP region. This suggests that closed MCC clouds over the NP might be more influenced by static stability, which aligns with their larger EIS values, compared to open MCC clouds (Fig. 4 and Table 1).

The relationship between both large-scale meteorological variables ($M$ index and EIS) and each MCC category is graphically shown with two-dimensional histograms of $M$ versus EIS (Fig. 5). Each two-dimensional histogram is calculated using a composite of the MCC clouds identified for each region. The highest MCC frequencies occurs for $M$ index values between -2 and 5 K for both regions. The open MCC frequency in the NP is characterized by the highest $M$ index values with a difference of 3 K higher compared to the maximum frequency of open MCC over the SO, consistent with stronger MCAOs in the NH than in the SH (Fletcher et al., 2016a). Open MCC clouds also show negative values of EIS, suggesting a condition of atmospheric instability. Both increased surface forcing due to the warmer ocean-cooler air temperature contrast (positive $M$ index) and static instability (negative EIS) have previously been established as drivers of open MCC cloud development (McCoy et al., 2017). Closed MCCs have a weaker relationship with $M$, depending more strongly on EIS. Figure 4 shows that the most pronounced difference between open and closed MCC is found for EIS. The figure illustrates that closed MCC clouds are consistently linked to larger EIS values over the NP, while open MCC clouds display the opposite behavior. On average, EIS

for closed MCC is 5 K larger than open MCC values, ranging from about 5 to 10 K. For both open and closed MCC categories, the SO clouds have a better relation against EIS compared to the NP.

## 3.2 Summer and Winter Seasons

220 As previously discussed, these mid-latitude storm tracks undergo a strong seasonal cycle, particularly across the NP where the storm track largely collapses over the summer (Hoskins and Hodges, 2005). As such, it is worthwhile to extend our analysis to the winter and summer seasons, separately (Fig. 6). In general, open MCC exhibits a considerable seasonal difference between summer and winter. In both regions, the maximum frequency of occurrence for open MCC clouds is found during the winter season, with values of 40 % for the NP and 27 % for the SO. During summertime, a considerable reduction in the frequency of 225 occurrence for open MCCs is observed for both regions, especially for the NP, with maximum values of 13 % and 20 % (NP and SO, respectively). This strong seasonality in the open MCC frequency can be related to the frequency of occurrence of MCAOs as observed in Figures 2e,h. The presence of cold air outbreaks is considerably higher during wintertime, especially for the NP, reaching frequencies of occurrence of 90 % over this region. Figure 2e shows that during summer it is still possible to detect MCAOs over the SO, in contrast to the NP.

230 Compared to open MCC, the frequency of closed MCC shows less interseasonal variability (Fig. 6a,c), with peaks slightly higher during summer, 15 % and 13 % for the NP and SO, respectively. During this season, the storm track shifts poleward in both hemispheres when baroclinicity is weakest, typically ranging from around 40 to 60° (Shaw et al., 2016). Consequently, the frequency peaks of open MCCs move poleward along with the storm track. For both seasons, the southeastern Indian Ocean (SO region) and east of Japan (NP region) are the areas with the highest frequency of occurrences for closed MCCs. 235 The seasonal two-dimensional histograms of $M$ versus EIS for open and closed MCC categories are shown in Fig. 7. While the presence of MCAOs is also a factor in the development of closed MCC, the seasonal spatial distribution of the EIS shown in Fig. 2d,g displays a better alignment with the peaks in both hemispheres and for both seasons. This is consistent with the strong correlation between the EIS and closed MCCs (Figures S4 and S5 in Supplementary Material), showing that the EIS is a more relevant factor for development of closed MCCs. The distribution of the summer peaks of MCC closed clouds occurrence 240 for both hemisphere is correlated with higher EIS (Fig. S4), which is consistent with Fig. 7, where high EIS are associated with higher presence of closed MCC clouds.

The seasonal differences in both EIS and $M$ indices and their influence on MCC clouds during the same season, either winter or summer, may largely be explained by the large-scale dynamic, thermodynamic influences of the SST and the fact that land masses represent a smaller surface area in the SO compared to the NP. The NP has a large seasonal cycle in SST (Figures 2g,i), 245 particularly at the high latitudes, where the ocean basin is relatively shallow (Sea of Okhotsk and the Bering Sea). This implies a significant warming during the summer months, which leads to a notable seasonal variation in SST. This contrasts sharply with the high latitudes of the SO, where SST experiences a relatively subdued seasonal cycle. Such substantial SST seasonality over the NP is a key driver behind the pronounced seasonal variation in MCAO events, as categorized by Fletcher et al. (2016a). Moreover, this strong SST seasonality is a contributing factor to the seasonal disappearance of the NP storm track (Wu and 250 Kinter III, 2010). For both hemispheres, the winter season exhibits the extension of colder SST towards the equator and stronger

gradients (Fig. S1). During this season, the frequency of occurrence of open MCC clouds is higher compared to summer. While the frequency of occurrences of closed MCCs is higher at lower latitudes than at high latitudes during wintertime.

## 3.3 Diurnal Cycle

The diurnal cycle of the MCC frequencies is calculated over boxes of $10 \times 10°$, center at $40°$ N, $170°$ W (NP) and $45°$ S, $130°$ W (SO) (Fig. 8). Both boxes represented areas with high frequencies of MCC clouds. Comparing the annual means between NP and SO, closed MCC over the SO exhibits the most pronounced daily cycle with higher frequencies at night, and a peak of 13 % at 07:00 local standard time (LST) and a minimum at 15:00 LST (6.2 %, Fig. 8b), with a range of the cycle of $\sim 7$ %. In contrast, closed MCC over the NP exhibits a less noticeable diurnal cycle, with a peak of 21 % at 02:00 LST and a range of the cycle of $\sim 4$ % (Fig. 8a). It is also noticeable that the standard deviation of the open MCC frequency shows a higher variability compared to closed MCC clouds over the SO, while the standard deviations for open and closed MCC frequencies show similar variability over the NP. Looking at the summer and winter means, a closed MCC diurnal cycle was identifiable for both seasons over the SO, being most intense during the summer months (December–February), with a range of the cycle of $\sim$ 7 %. During wintertime, a closed MCC diurnal cycle in the NP is identifiable as well, while a diurnal cycle in the summertime NP (June-August) is almost absent. For closed MCCs, the seasonal standard deviations show larger differences between both seasons, with a lower variability during summer and higher during winter for both regions. In contrast, the seasonal standard deviations of the open MCC frequency is lower during summer over the NP and the SO. A contrast between both regions is that the most pronounced daily cycle for closed MCCs is in opposite season, summer over the SO ($\sim 4$ % of diurnal range), and winter over the NP ($\sim 5$ % of diurnal range). Diurnal cycles for $M$ index, EIS, SST and near-surface wind speed were examined, but no discernible signal or pattern was found throughout the cycle (Figures S6-S9 in Supplementary Material).

The annual diurnal cycle of open MCC is less distinct for both regions, with maximums of 29 % at 11:00 LST and 23 % at 21:00 LTS, NP and SO respectively. The standard deviation for open MCC shows a large variability during the day for both regions. Compared to the SO closed MCC, open MCC shows more variability throughout the day. During wintertime, the open MCC occurrence over the NP shows a slightly more noticeable daily cycle with a peak of 38 % at 11:00 LTS (Fig. 8e) and with the highest variability for both regions and seasons ($\sim 13$ %). The open MCC occurrence shows higher frequencies and variability during winter that summer in both regions.

## 3.4 Low-level clouds within the category 'Other'

The annual frequency of occurrence results for the subcategory Other low-level clouds (LLC) are depicted in Fig. 9a. For open and closed MCC clouds distributions, the results for only daytime observations remains consistent with the comprehensive 24-hour observational record (not shown). Focusing on the other LLC here, we find that in the NP, maxima are located south of $30°$ N with a peak of 51%. Conversely, peaks in the SH are more pronounced, reaching up to 65%. Overall, the SO displays a greater frequency of occurrence for other LLC. Notable peaks are found poleward of $50°$ S over the SO and in the vicinity of Western Australia in the Southeast Indian Ocean. This aligns with prior observations where the SO has been found to have

a higher frequency of low-level clouds (e.g., Mace et al., 2009; Muhlbauer et al., 2014). Observations indicate a prevalence of approximately 90% of low-altitude clouds over the SO (Huang et al., 2012b).

Table 2 presents the annual mean values for individual cloud properties for the three categories observed during daytime: open MCC, closed MCC, and other LLC. When comparing cloud properties between the NP and SO regions, the most notice-able differences are observed in cloud optical thickness and cloud effective radius. Over the NP, the cloud optical thickness is higher for all the categories, this suggests a couple of key differences. Firstly, clouds in the NP region are likely to have higher reflectivity or albedo, potentially exerting a cooling effect on the Earth's surface. Secondly, these optically thicker clouds may also be more effective at trapping outgoing longwave radiation, potentially leading to a warming effect on the surface below. The cloud effective radius is greater over the SO compared to the NP, suggesting the presence of larger cloud droplets in the SO region.

In general, other LCC displays a notable seasonal variation between summer and winter (Figures 9b,c). The seasonal cycle is more pronounced over the NP, largely due to the marked seasonality in factors controlling low-level clouds, such as SST (Fig. 2). Conversely, in the mid-latitude storm track regions where both MCC categories appear more frequently, especially open MCC clouds, the seasonal cycle of other LLC is less pronounced (see Fig. 6). For both regions, the peak occurrence of this subcategory is observed during the summer season, registering values of 75% in the NP and 73% in the SO. While the winter peaks correspond to 45% for NP and 58% for SO, respectively.

## 4   Discussion and conclusions

In Lang et al. (2022), a CNN model was developed to identify and classify open and closed MCC clouds from high-frequency geostationary Himawari-8 satellite observations over the SO. Here, this analysis is extended to the North Pacific to identify differences in the organization of MABL clouds between the hemispheres. The inputs to the CNN model consist of hourly brightness temperature from AHI Himawari-8 during the period 2016-2018 for the regions defined as SO (20-60° S, 80°E-160° W) and NP (20-60° N, 80°E-160° W). The use of high-resolution geostationary Himawari-8 satellite data, rather than polar-orbiting satellite data, offers the advantage of providing more robust statistics for analysis. Different large-scale meteorological variables were chosen to explore their impact on MCC cloud distributions.

The climatology for both regions showed that MCC clouds are roughly distributed over the mid-latitude storm tracks of both hemispheres, with peaks poleward of the 40° latitude. The distribution of MCC clouds aligns with the spatial pattern of shallow clouds that can be commonly found in the mid-latitude as observed by CloudSat/CALIPSO (Muhlbauer et al., 2014). and MODIS (McCoy et al., 2017, 2023). Open MCC is more prevalent than closed MCC in both regions, and the highest frequencies are observed over the SO. The local frequency maximums of open MCC are 25% for the SO and 16% for the NP. Closed MCCs are characterized by lower frequencies, which are relatively comparable between both regions. The SO and the NP exhibit local maximums of 12% and 14%, respectively, for closed MCC occurrences. The annual mean frequencies in Fig. 2b show that closed MCCs are more frequently found at higher latitudes over the NP, consistent with Rampal and Davies (2020), whereas these clouds are less prevalent at high latitudes of the SO. Truong et al. (2020) found that across the SO storm

track and higher latitudes, the presence of multilayer clouds is a common feature, which might explain a low frequency of occurrence for SO closed MCCs (Mace et al., 2009).

Previous studies have linked the MCAOs to the presence of open and closed MCC clouds over mid-latitude (e.g., McCoy et al., 2017). This is mainly due to the enhanced surface forcing when these cold air masses are advected over warmer oceans, which increases the turbulent heat and moisture (i.e., latent plus sensible heat flux) fluxes from the surface into the marine boundary layer, driving the formation of MABL clouds (Abel et al., 2017; Fletcher et al., 2016b; Kolstad et al., 2009). The frequency of occurrence of MCAOs for the analyses period is higher over the NP, with a peak frequency of occurrence of around $\sim 50\%$ in some areas. Notably, the Kuroshio region and its western boundary current stand out as areas with particularly high frequencies of MCAOs (Fig. 2b). Over this region, open MCC clouds were often associated with stronger cold air outbreaks as measured by $M$ index value (Figures 2 and 4), with values on average 3 K higher for open MCCs that in the SO (Fig. 5). Considering the spatial relationship between the frequency of open and closed MCC against the EIS, $M$ index, SST and near-surface winds, the correlation coefficient maps (Fig. 4) suggest that closed MCC clouds in the NP might be more sensitive to static stability. The stronger correlations between closed MCC and EIS over the NP compared to the SO are in line with the results reported by Muhlbauer et al. (2014), who found a correlation coefficient of 0.8 for the NP and only 0.2 for the SO. Over the SO storm track, there is limited statistically significant correlation for closed MCC with most forcing factors. Closed MCCs demonstrate a stronger correlation with SST off the coast of Western Australia. This relationship suggests that the cold current in the region may have a role, with the advected warm air contributing to the formation of MABL clouds. Another distinction between the two regions is also observed, the SO closed MCCs are associated with higher static stability compared with the NP region (Fig. 5), which is again consistent with Muhlbauer et al. (2014), where they found lower annual means of EIS for the NP. For open MCC clouds, Fig. 4 displays stronger correlations with near-surface wind speed and the $M$ index. A robust association between open MCC clouds and MCAOs has been previously documented in McCoy et al. (2017), corroborating the higher occurrence of MCAOs as shown in Fig. 2. In relation to the association between open MCCs and near-surface wind speed, previous studies have shown that open MCC cloud conditions tend to coincide with windier environments (Jensen et al., 2021; Wood et al., 2008).

In the subtropics, the transition from closed to open MCC clouds occurs under relatively homogeneous meteorological conditions, where strong winds precede intense drizzle, leading to the transition (e.g., Eastman et al., 2022; Yamaguchi et al., 2017). Drizzle is transported into the stratocumulus layer by turbulence, where it intensifies and breaks up the stratocumulus clouds, removing aerosols and inducing a positive feedback loop that accelerates the transition (Yamaguchi et al., 2017). Precipitation also plays a significant role in cloud morphology transitions from closed to open clouds in the mid-latitudes during MCAOs, driven by a decoupling of the boundary layer induced by precipitation (Abel et al., 2017; Tornow et al., 2021). In situ observations have revealed that the transition from closed MCC clouds to open MCC clouds during MCAOs in the SO (Lang et al., 2021) and the North Atlantic (Abel et al., 2017) frequently includes mixed-phase and supercooled clouds. Recent in-situ observations from SO field campaigns further suggest the commonplace occurrence of secondary ice production, such as through the Hallett-Mossop process, in the open MCC clouds, which is linked to enhanced precipitation production (Huang et al., 2021; Lasher-Trapp et al., 2021; Järvinen et al., 2023).

The seasonality of open MCC clouds (Fig. 6) is particularly strong, consistent with their relationship to MCAOs (Figures 2 and S4). In general, the frequency of open MCCs is higher over the NP than their SO counterparts during the respective winter. This larger seasonal difference between the NP and SO regions is evident for both open and closed MCC clouds, and it is closely related to the SST gradients, as shown in Fig. S1. During winter, the mean SSTs in the NH are notably colder compared to those in summer, resulting in a more pronounced seasonal variation in the NH when compared to the SH. This observation indicates that SST gradients play a significant role in influencing the seasonal differences in MCC cloud patterns between the two study regions. In Addition, the difference in the landmass between both regions also influences the ocean-atmosphere interaction. The spatial distribution of the $M$ index (Fig. 2) does not align strongly with the maximum occurrence of open MCC, especially over the NP during winter. The $M$ index peaks closer to the Asian land mass. An explanation might be that very strong air-sea temperature difference (high $M$ index values) leads to rapid convection that does not necessarily produce well-formed open cells. Despite this spatial misalignment, the two-dimensional composite histograms [both annual (Fig. 4) and by season (Fig. 7)] suggest a relationship between them. On the other hand, the seasonal differences in the EIS are consistent with Muhlbauer et al. (2014), where the annual cycle is found to be stronger over the NP as observed in Fig. 2. During both seasons, closed MCC clouds are positively correlated with EIS compared with open MCC clouds (Figures S4 and S5). This association between closed MCCs and the static stability is also observed in the spatial distribution throughout the year over both regions, the maximum of closed MCC clouds frequencies is strongly correlated to the EIS.

Taking advantage of high-temporal resolution Himawari-8 observations, we explore the diurnal cycle of both MCC morphologies over both regions, which has shown that the frequency of occurrence of closed MCC exhibits a distinctive daily cycle over the SO, while the daily cycle of closed MCCs in the NP is less noticeable. A nighttime maximum for closed MCC clouds has been particularly associated with the absence of solar forcing, allowing the marine boundary layer to become well mixed due to the buoyancy driven by the cloud radiative cooling, and the cloud deck commonly thickens with the renewed access to moisture from the ocean surface (Minnis and Harrison, 1984; Nicholls, 1984). We also observed that the daily cycles have a maximum amplitude in different seasons depending on the region, over the SO the most pronounced daily cycle for closed MCCs is during warmer months, while over the NP is during winter. We speculate that the differences in the diurnal cycles of open and closed MCC clouds between the SO and the NP might be attributed to several factors. These include differences in solar radiation due to their hemispheric locations and timing, variations in ocean currents and sea surface temperatures, the influence of surrounding landmasses, and differences in atmospheric stability. For instance, the study found that closed MCC clouds in the NP are notably sensitive to static stability, as indicated by higher EIS values. Additionally, the most pronounced daily cycle for closed MCCs occurs in opposite seasons in the two regions, with summer over the SO and winter over the NP, suggesting that seasonal variations also play a significant role. These factors are complex and interrelated, and understanding their precise influence would require further detailed study.

Future work could involve conducting a sensitivity analysis of various machine learning techniques, similar to CNN models, for detecting spatial patterns. Additionally, exploring the application of other types of neural networks that incorporate spatial autoregressive models and geostatistical methods could be beneficial for identifying spatial relationships of MCC cloud types. Furthermore, by incorporating additional categories, such as disorganized MCC, no MCC, and low-level cloud categories as

defined by Wood and Hartmann (2006), it is possible to develop a more comprehensive understanding of the marine atmospheric boundary layer clouds. In conjunction with our exploration of the influence of large-scale environmental factors like $M$ and EIS, it is of importance to examine how the spatial organization of MCC clouds influences the daily cycle of shallow precipitation.

390 *Code availability.* The code related to this article is available online at: https://doi.org/10.5281/zenodo.5657269

*Data availability.* All Himawari-8 data can be accessed using the following public website: https://www.eorc.jaxa.jp/ptree/index. html (Japan Meteorological Agency, 2023).

*Author contributions.* FL performed the data analysis and wrote the article with the support of SS. FL and LA implemented the method to train the network model. All authors contributed to the discussion of the results and editing of the manuscript.

395 *Competing interests.* The authors declare that they have no conflict of interest.

*Acknowledgements.* Steven Siems is supported by Securing Antarctica's Environmental Future (SAEF), a Special Research Initiative of the Australian Research Council (SRI20010005).

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

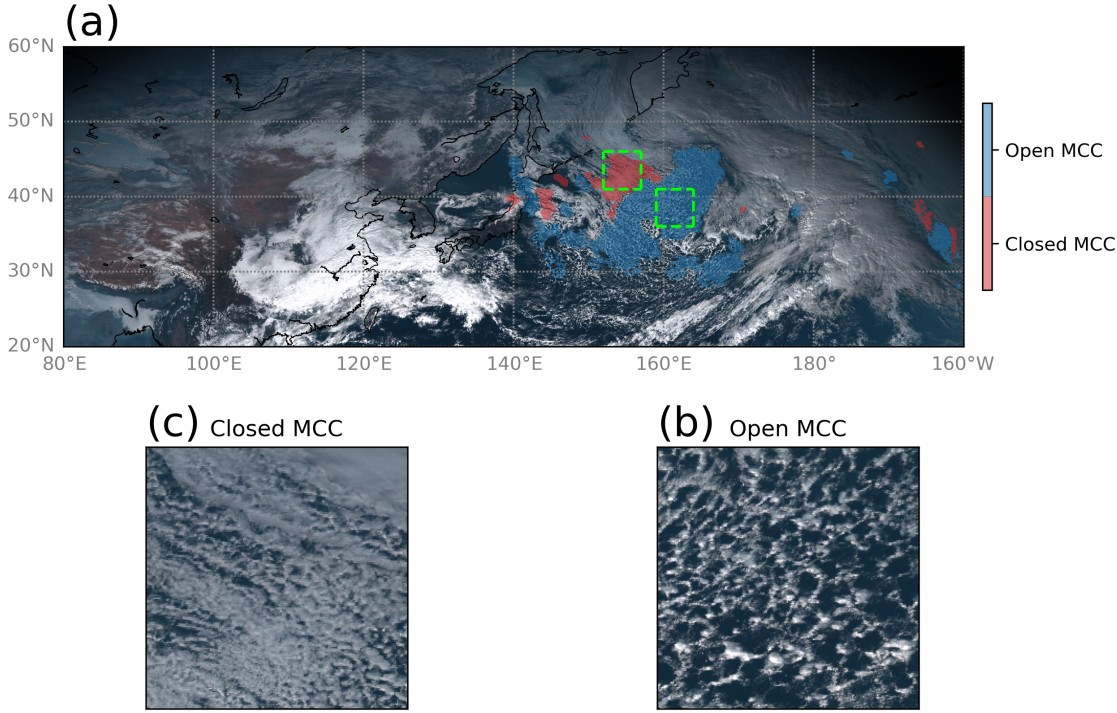

**Figure 1.** (a) An example scene of AHI Himawari-8 (visible) and MCC structures identified by the CNN on 7th January 2017 at 03:00 UTC. The green squares delimit the magnified area for the subscenes of 5° x 5° for (b) closed and (c) open MCC clouds.

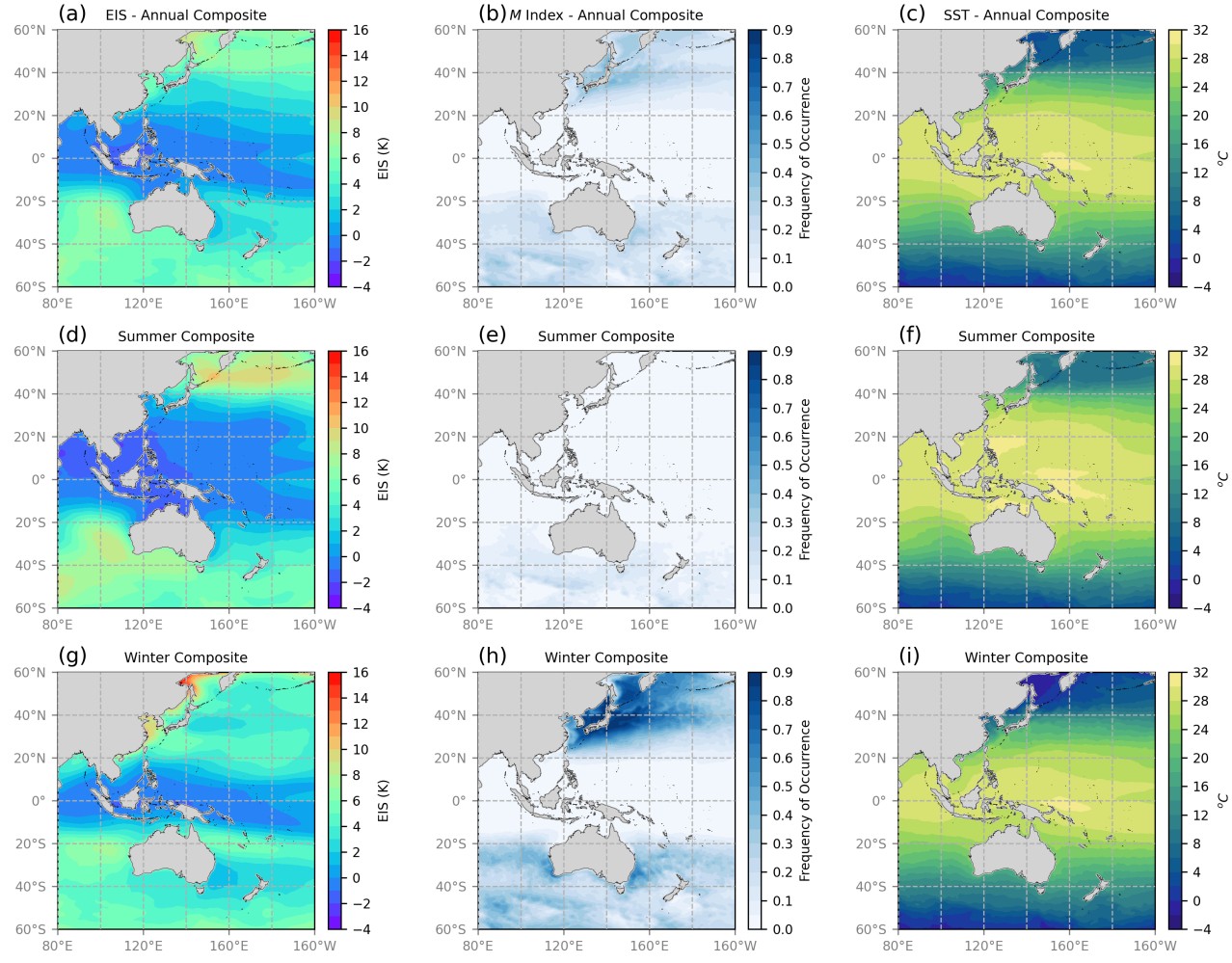

**Figure 2.** Annual mean (a-c), summer (d-f) and winter (g-i) seasons (2016-2018) for EIS, frequency of occurrence of $M$ index for cases where $M >0$ K, and mean SST from ERA5 reanalysis products. Austral (boreal) summer is defined by December-February (June-August), while austral (boreal) winter is defined by June-August (December-February).

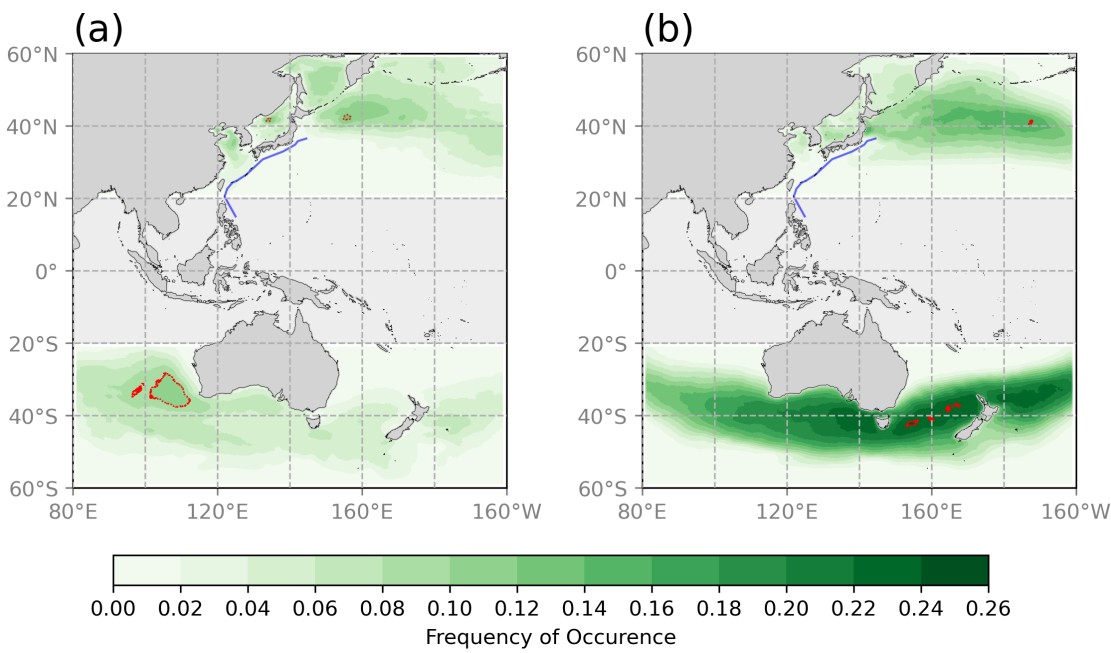

**Figure 3.** Distribution of the annual frequency of occurrence of the MCC clouds for the period 2016-2018. (a) Closed MCC and (b) Open MCC structures.The dotted red lines show the contour level of maximum frequencies for each region, and the blue lines indicates the approximate location of the Kuroshio current (Shen et al., 2022).

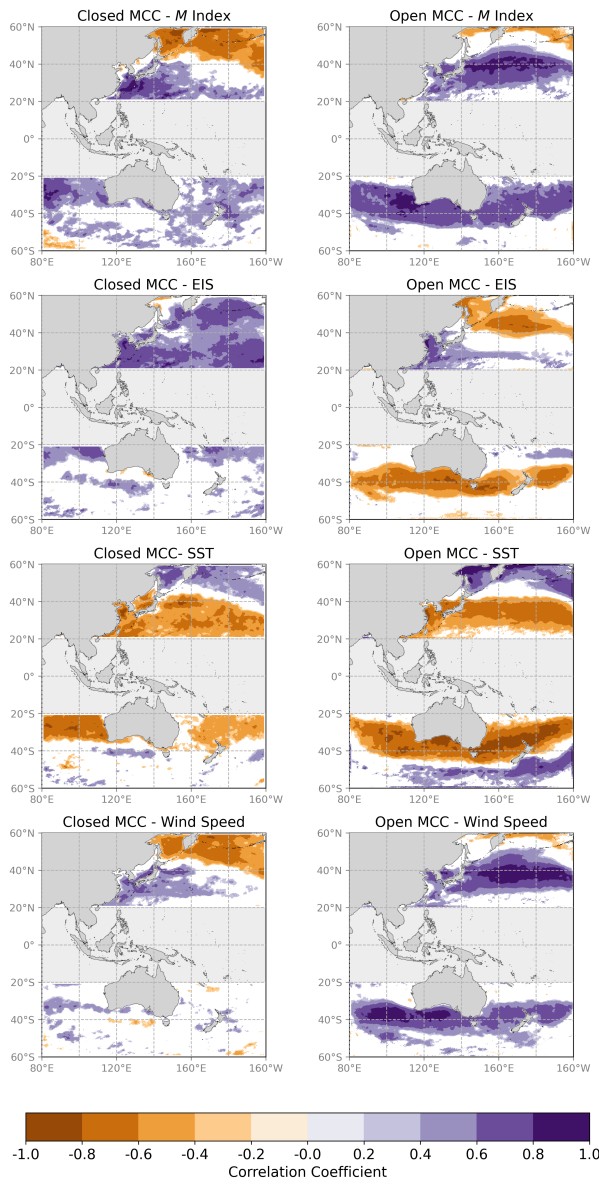

**Figure 4.** Correlation coefficients between both closed and open MCC cloud monthly occurrence frequencies and $M$ index, EIS, SST and near-surface wind speed from ERA5 reanalysis products. White areas indicates where results are not significant at a 95% confidence level.

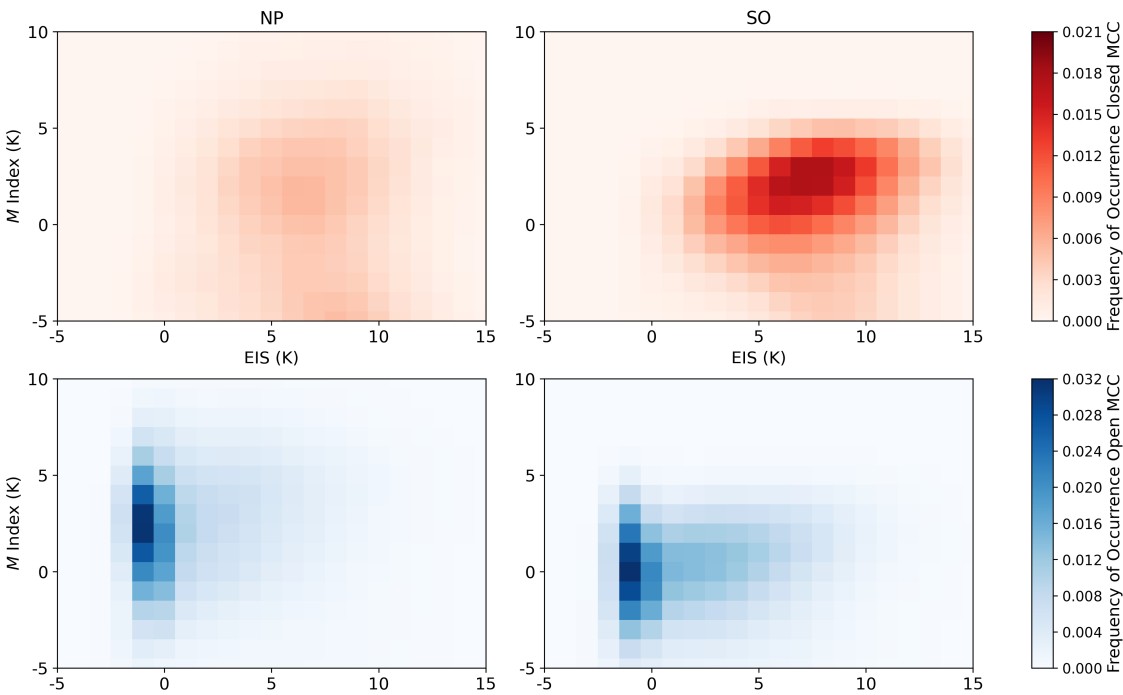

**Figure 5.** Two-dimensional composite histograms of $M$ index and EIS for closed (top row) and open (bottom row) MCC clouds for the period 2016-2018. Data is for both NP (left column) and SO (right column) regions. Both the EIS and $M$ index are calculated using a filter derived from the composite of MCC clouds identified for each region.

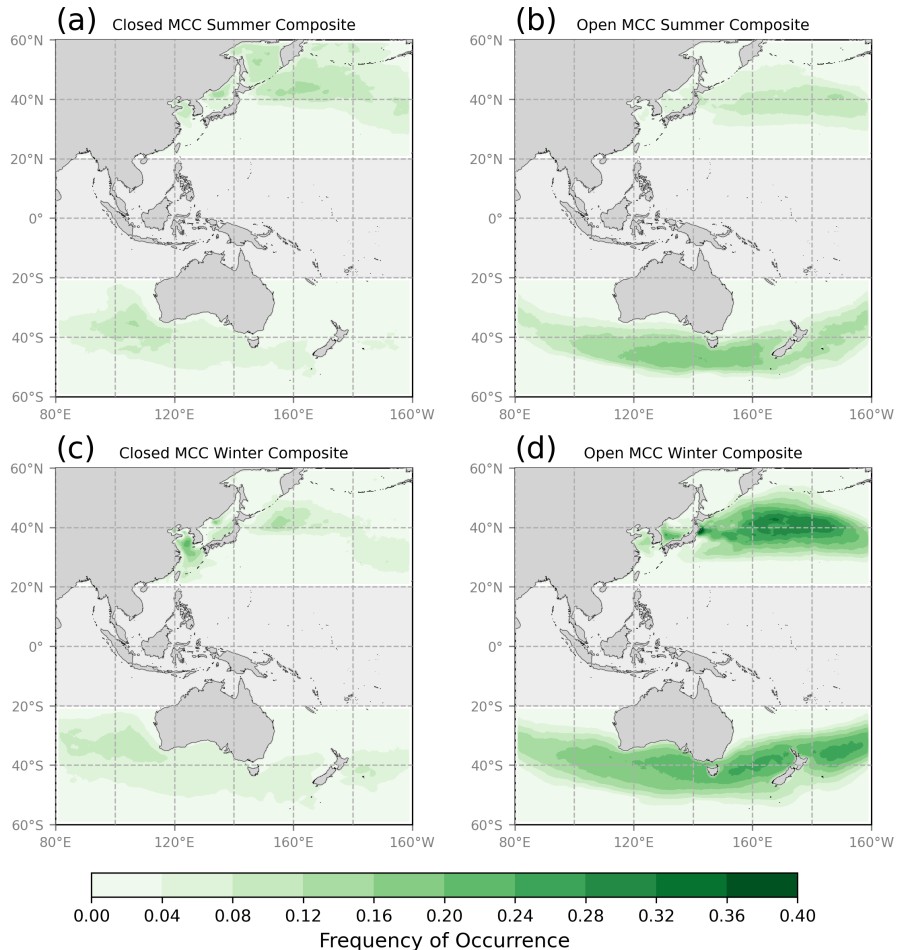

**Figure 6.** Summer and winter composites of the frequency of occurrence of MCC clouds for the period 2016-2018. (a,b) Boreal and austral summer together on the same plots and (c,d) boreal and austral winter together on the same plots. Austral (boreal) summer is defined by December-February (June-August), while austral (boreal) winter is defined by June-August (December-February).

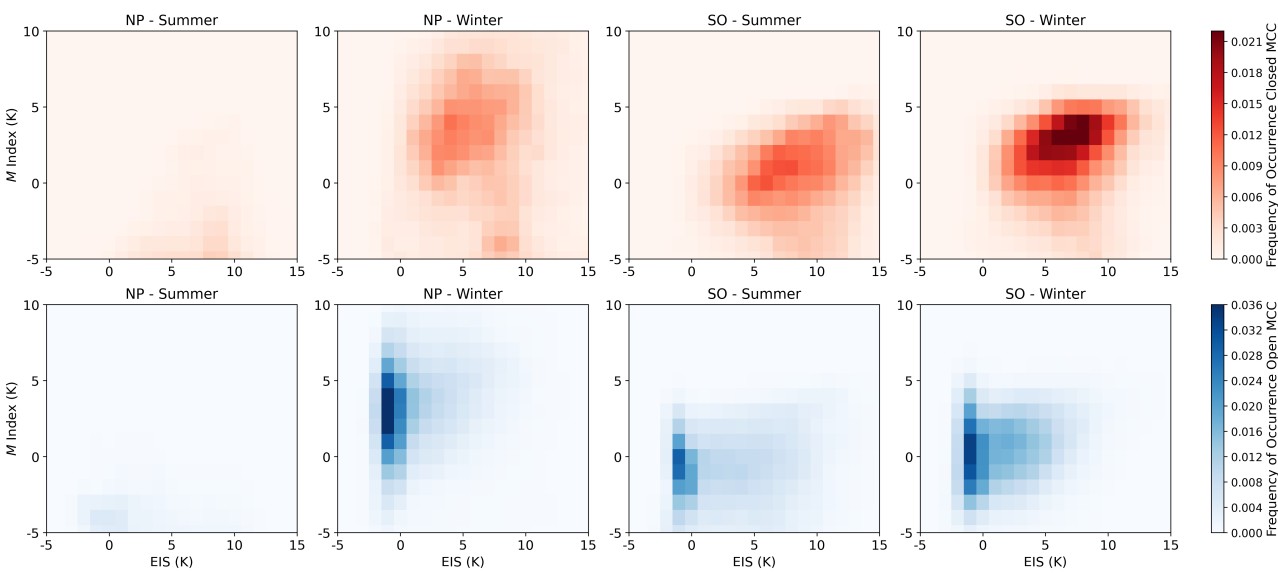

**Figure 7.** Seasonal two-dimensional composite histograms of $M$ index and EIS for closed and open MCC clouds for the period 2016-2018. Data is for both NP (left two columns) and SO (right two columns) regions. Austral (boreal) summer is defined by December-February (June-August), while austral (boreal) winter is defined by June-August (December-February). Both the EIS and $M$ indices are calculated using a filter derived from the composite of MCC clouds identified for each region.

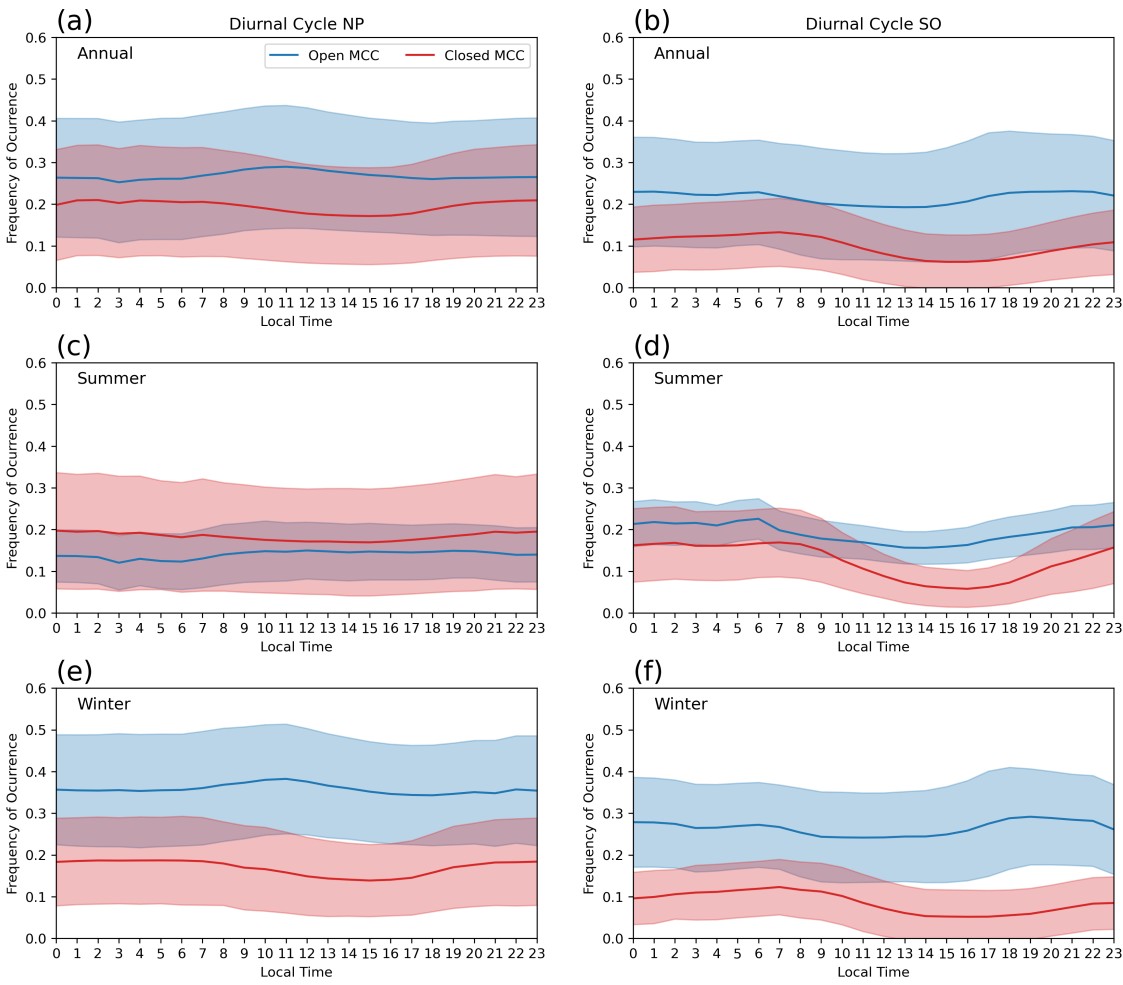

**Figure 8.** Diurnal cycle of the frequency of occurrence of MCC structures for the period 2016–2018. Shown are open MCC (blue) and closed MCC (red) structures. The diurnal cycle is calculated over boxes of $10 \times 10°$, center at $40°$N, $170°$W and $45°$S, $130°$W. Seasonal means are shown for summer and winter. Shadings represent 1 standard deviation. Frequencies are calculated for the latitudinal band between 40 and $50°$ for both hemispheres.

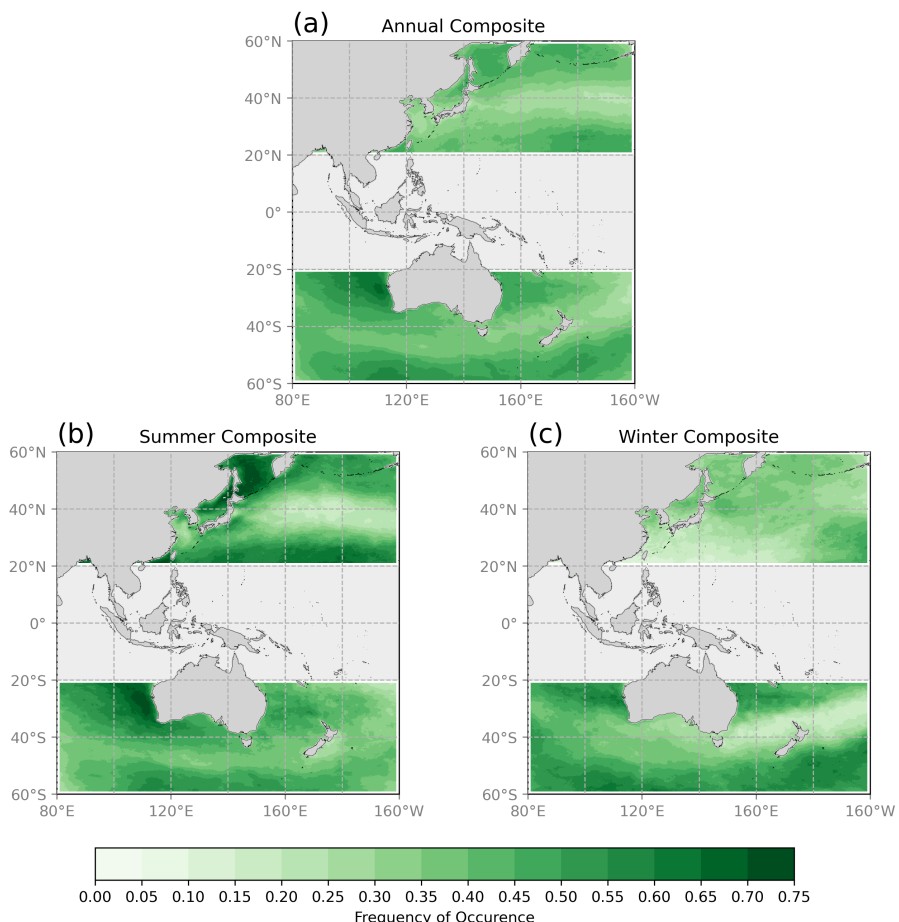

**Figure 9.** Distribution of the annual frequency of occurrence of the subcategory Other LLC for the period 2016-2018 during daytime. (a) Annual, (b) summer and (c) winter. Austral (boreal) summer is defined by December-February (June-August), while austral (boreal) winter is defined by June-August (December-February).

**Table 1.** Annual and seasonal means (standard deviation) of estimated inversion strength (EIS) and $M$ index. Austral (boreal) summer is defined by December-February (June-August), while austral (boreal) winter is defined by June-August (December-February). Both the EIS and $M$ indices are calculated using a filter derived from the composite of MCC clouds identified for each region.

| Region | Season | Closed MCC | | Open MCC | |
|--------|--------|------------|-------------|----------|-------------|
| | | EIS (K) | $M$ Index (K) | EIS (K) | $M$ Index (K) |
| NP | Annual | 7.8 (3.5) | -4.9 (7.6) | 2.6 (3.8) | 0.1 (6.0) |
| | Summer | 9.0 (1.8) | -11.4 (2.5) | 7.7 (1.8) | -11.9 (2.3) |
| | Winter | 6.9 (2.2) | 3.1 (2.6) | 2.3 (1.5) | 3.5 (1.3) |
| SO | Annual | 7.5 (3.2) | -0.1 (3.8) | 2.6 (3.5) | -0.5 (3.3) |
| | Summer | 8.6 (1.9) | -2.4 (2.5) | 4.7 (1.6) | -2.5 (1.8) |
| | Winter | 7.1 (1.5) | 1.6 (1.2) | 2.6 (0.9) | 0.9 (0.9) |

**Table 2.** Annual means (standard deviation) of cloud-top height, cloud optical thickness, cloud-top temperature, and effective radius for both regions and for the three categories (daytime observations).

| Region | Category | Cloud-top height (km) | Cloud optical thickness | Cloud-top temperature (K) | Cloud effective radius (µm) |
|--------|----------|-----------------------|-------------------------|---------------------------|------------------------------|
| NP | Closed MCC | 2.73 (0.09) | 18.18 (1.97) | 271.8 (0.8) | 13.50 (0.82) |
| | Open MCC | 2.55 (0.15) | 13.12 (2.00) | 269.6 (0.8) | 21.40 (1.30) |
| | Other LLC | 2.30 (0.08) | 13.07 (2.01) | 274.7 (1.1) | 18.26 (1.17) |
| SO | Closed MCC | 2.62 (0.08) | 15.7 (1.60) | 271.7 (0.7) | 15.61 (0.73) |
| | Open MCC | 2.05 (0.15) | 9.73 (1.62) | 269.8 (0.8) | 25.26 (1.20) |
| | Other LLC | 2.22 (0.09) | 11.6 (1.42) | 272.7 (1.0) | 19.83 (1.01) |