# Peer review of "On the relationship between mesoscale cellular convection and meteorological forcing: Comparing the Southern Ocean against the North Pacific"

_EGUsphere, 2023_

## Author Comment (AC1)

**Review report**

The authors investigate the connection between meteorological parameters and the morphology of low-level clouds in two regions: a portion of the Southern Ocean and the North Pacific. To identify three morphological classes (i.e., "closen", "open", and "other") the authors apply a neural network that was prepared in a previous publication (Lang et al., 2022) to radiance fields from the geostationary HIMAWARI satellite. The article documents the cooccurrence of class frequency with average estimated inversion strength (EIS) and a marine cold-air outbreak index, M, mainly finding that closed cells are connected to greater EIS.

The article contributes to a topic that is of great relevance to the climate community. The authors importantly leverage imagery from a geostationary platform to identify cloud morphology on a great temporal resolution – a scientific advancement that was achieved in Lang et al. (2022). With respect to the previous paper, the present article falls short of making a substantial contribution. Below, I'm outlining major issues that lead me to recommend a rejection but encourage a resubmission in improved form. In a nutshell, the authors should (1) include statistics from the class "other", (2) quantify their results, (3) contextualize the selected meteorology parameters and possibly widen the parameter space, and (4) inspect EIS throughout the text, table, and figures.

**Major issues**

(1) The classification resulted in peak frequencies of ~20% for "open cells" and ~15% for "closed cells", leaving at least ~65% of the samples classified as "other". The authors explain that "other" constitute cases of mid- and high-level clouds, but also low-level clouds that were different enough from the first two classes (ll. 94-96). Figure 1 shows a classification example in which apparent open cellular cloud decks (e.g., ~35degN, ~160degE) were classified neither as "open" nor as "closed" (leading me to believe they were joined "other" which is not shown here). Given that "other" makes up the bulk of the samples and that the nature of these samples is unknown to the reader, the authors need to (1) provide information on them, showing, for example, statistics of cloud-top height, cloud fraction, and cloud optical depth of "other" compared to "open" and "closed" and (2) include "other" where currently only "closed" and "open" is shown. Both aspects will become highly relevant for the diurnal cycle that shows a decrease of "closed" in the afternoon but no increase of "open" at the same time, leading to believe that the class "other" increased in the afternoon and begging the question why that is (e.g., truly a transition from "closed" to "disorganized" or an increase in local cirrus cloud fraction?).

R: Thank you for your comment. We acknowledge the importance of analyzing low-level clouds within the category "Other". However, it is important to note that in our study this category indeed encompasses a diverse range of covers, including not only other types of clouds but also land or ocean covers as seen from Himawari-8. As we discussed in Lang et al. (2022), one of the limitations of our study is that we did not include other low-level cloud types, such as disorganized MCC and no MCC clouds. To address this limitation and improve our Convolutional Neural Network (CNN), we plan to incorporate more training samples from both regions to include these additional categories.

Regarding the classification of open MCCs in Figure 1, in Lang et al. (2022), we openly acknowledged the uncertainties associated with using a CCN technique in the separation of disorganized and open MCCs. This uncertainty was also reported by Yuan et al. (2020). We understand the significance of refining the classification methods, and we appreciate your valuable input, which has been taken into account for future improvements in our research.

On the other hand, the Japan Aerospace Exploration Agency (JAXA) provides Himawari-8 cloud products such as cloud-top height and cloud optical thickness. However, these data are not available during nighttime. Therefore, in our first version of the manuscript, we made the decision not to use these products to calculate statistics related to the three categories since we lacked continuous access to the products 24 hours a day. This limitation of the Himawari-8 cloud products hinders the estimation and analysis of the diurnal cycle for the category "Others". Nonetheless, we have addressed this limitation by extending our analysis to include the low-level clouds within the category "Others" and comparing them with the open and closed MCC categories using the Himawari-8 cloud products. To define the low-level clouds in the category "Others," we applied a filter considering only the diurnal clouds product, identifying clouds below 3.5 km, and selecting daytime open and closed MCC. The results of this analysis are presented in Section 3.4, where we examine and compare the characteristics of these low-level clouds with open and closed MCC. Additionally, we provide statistics of the cloud products available on the JAXA server for the three categories. This additional analysis allows for a more comprehensive understanding of the low-level cloud characteristics in the "Others" category and their relationship with open and closed MCC.

(2) The authors compare class frequency and meteorological fields and often speak of "correlation" (e.g., ll. 161-164, l. 194, ll. 254-255) or even "high correlation" (ll. 267-268) when visually comparing maps. However, there is no actual calculation of correlation. The paper would benefit from a more quantitative rather than qualitative analysis. Given the great temporal resolution, the authors could not only compare seasonally averaged maps but actually compute correlation on a pixel-by-pixel basis using the native 10-min resolution.

R: We thank you for this comment – we agree. In the revised manuscript, we have calculated correlation maps (new Figure 4) to ensure consistency in assessing the correlation between both categories of MCC and the meteorological indices $M$ and EIS. We have also extended our analysis to include correlation maps for near-surface wind speed and sea surface temperature (SST). These correlation maps provide a quantitative analysis of the associations between monthly MCC cloud frequency and monthly averages of $M$ index, EIS, SST, and near-surface wind speed in both study regions. The correlations were computed using the monthly mean MCC cloud frequencies in each grid box, and they demonstrate statistical significance at a 95% confidence level for a 36-point correlation. Based on this correlation analysis, we have diligently revised the manuscript to include the quantitative analysis in all sections and paragraphs as suggested by the referee. The inclusion of these correlation maps has provided us with deeper insights into the relationships between the meteorological variables and both open and closed MCC cloud types, thereby enhancing our understanding of their spatial patterns and behavior. We believe that these additional analyses strengthen the validity and significance of our findings, and we are grateful for the referee's valuable input in improving the quality of our research.

(3) The authors selected EIS and the M index as meteorological parameters, and it is unclear why they selected these two and why the analysis excludes other parameters. For example, near-surface wind speed was recently connected to cloud morphological transitions in Eastman et al. (2023). Also, the authors examine the diurnal cycle of class frequency but leave out diurnal cycles of meteorological parameters that were used for the seasonal analysis a few pages before.

R: The selection of both meteorological indices is well-founded on previous studies where the relationship of these indices with MCC clouds has been demonstrated. However, in response to the referee's suggestion, we have performed an additional analysis of near-surface wind speed and its relationship with both open and closed MCC cloud types as shown by the correlation maps in Figure 4 in the revised manuscript. By including this new analysis, we aim to provide a more comprehensive examination of the factors influencing the spatial patterns and behavior of both MCC clouds, thereby enriching the overall findings of our study. We are appreciative of the referee's valuable input, which has led to further improvement and robustness of our research.

To address the absence of the diurnal cycle for the meteorological parameters in our study, we have now included the diurnal cycle of the four variables considered in the correlation maps ($M$ index, EIS, SST, and near-surface wind speed) in the Supplementary Materials. These results indicate that no distinct signal or consistent pattern was observed throughout the cycle.

(4) The authors equate EIS with static stability in this paper (e.g., l. 11, 164), even though it was introduced as an indicator of which (ll. 110-111). Given that sea-surface temperature can exceed lower-tropospheric air temperature (giving positive M but possibly negative LTS), EIS can become negative, as the authors show in Figure 4. The authors should explain what a negative EIS means and whether EIS still holds as an indicator of static stability in cold-air outbreak conditions. In addition to revising EIS language, there seem to be inconsistencies across Figure 4, Figure 2, and Table 1: Figure 4 shows mostly negative EIS, yet in Figure 2 and Table 1 EIS is shown to be positive. The authors need to explain why that is (e.g., do composites in Figure 2 and statistics in Table 1 cover different time frames compared to Figure 4?).

R: Thank you for your insightful comment. In our revised manuscript, we have provided further clarification: while EIS is indeed an indicator of static stability, it should not be considered equivalent to it. We appreciate your bringing this to our attention.

Negative EIS can indeed occur when the SST exceeds the lower-tropospheric air temperature. This typically suggests a condition of atmospheric instability, where the temperature profile of the atmosphere does not inhibit vertical movement of air parcels. In such cases, EIS could become a less reliable indicator of static stability. It is been previously established by McCoy et al., (2017) that the development of open MCC clouds can be driven by two primary factors: enhanced surface forcing, stemming from the contrast between warmer ocean and cooler air temperatures (signified by a positive $M$ index), and static instability, indicated by negative EIS values. In the revised version of our manuscript, we have elaborated on the conditions associated with a positive $M$ index and negative EIS, as depicted in Figure 4 (now referred to as Figure 5).

Regarding the inconsistencies mentioned, we have double-checked our calculations. Figure 2 shows the mean values that are independent of the location of MCC clouds and cover the entire area, as defined in Section 2.3 (between 60°N-60°S and 80°E-160°W). In contrast, Figure 4 and Table 1 were calculated using a filter based on the location of open and closed MCC clouds for each time step from 2016 to 2018. We can confirm that both Figure 4 (new Figure 5) and Table 1 utilize the same data to compute the EIS values, and the results are presented as such. For further clarity, we have appended text to the captions of Figure 4 (new Figure 5) and 6 (new Figure 7). This addition to the captions and Table 1 explains that both the EIS and $M$ indices were determined using a filter derived from the MCC cloud locations and times, identified for each region.

**Minor issues**

ll. 10-11 This sentence is hard to understand. Please rephrase.

R: We have rewritten the sentence in the abstract to explain the difference in behavior between closed and open MCC clouds.

ll. 23-26 Please also cite a recent paper by McCoy et al., (2023).

R: This is a great suggestion. We have added McCoy et al., (2023) as reference in the examples of closed MCC clouds over the SO.

ll. 26-27 This sentence seems out of context.

R: Thank you noticing this sentence. We have removed this sentence in the revised manuscript.

ll. 95 Please clarify whether low-level clouds can be categorized as "other".

R: Thank you for this suggestion. We have indicated that the category called "Other" is used for all other coverage, including mid- and high-level clouds, as well as other types of low-level cloud types such no MCC, disorganized MCC clouds, and even clear skies. In addition, the analysis of a subcategory, which considers low-level clouds within the category "Other" is presented in Section 3.4.

Fig. 1 "Other" is included in the legend, but not shown in the map. Perhaps selected a color different from white.

R: The category "Other" encompasses all cover types other than open and closed MCC clouds, including oceans and land. To simplify the representation of the two main categories examined in this study, open and closed MCC, we have excluded the "Other" category from the legend.

Fig. 1 Please label the size of each box or add a reference bar to show distance.

R: Thank for this good idea. We have included the size of each box within the green squares in the caption of Figure 1.

Fig. 2 Please change label from "delta T (K)" to "EIS (K)".

R: Suggested revision made.

ll. 131ff. Would NP have great M values even when the Kuroshio current was absent? Also please quantify SST gradient.

R: The Kuroshio Current brings warm water from the tropics, so its absence or weakening can lead to a decrease in SST along the NP (Wang et al. 2022). Following the definition of the $M$ index in equation (2), at a lower SST, the values of the $M$ will decrease.

We have added a figure of the SST gradient as a Figure S1 in Supplementary Material.

ll. 141-142 To follow this sentence, please explain how cloud cover is related to both classes.

R: Thank you for this comment. We agree that the sentence was unclear. To clarify, we have added a sentence to explain that we performed visual inspections of several cases and time periods to confirm that our algorithm is capable of accurately classifying open and closed MCC clouds over the NP.

l. 145 Please substitute "total time" by "total number of observations".

R: Suggested revision made.

Fig. 3 Please mark maximum frequency to complement the text.

R: Suggested revision made.

Fig. 3 Please draw Kuroshio current in here.

R: Suggested revision made. We have added the Kuroshio current location to Figure 3.

ll. 150-151 (and also ll. 154-155 and ll. 174-175) It is unclear which class is referred to in this sentence.

R: We thank you for this comment. We have reorganized the paragraph to clarify the frequencies of occurrence and characteristics of both open and closed MCC categories.

l. 157 Please substitute "to the top" with "near the top".

R: Suggested revision made.

l. 160 Please substitute "SST gradient" with "average SST".

R: Suggested revision made.

l. 163 Please finish sentence; "as highly … as" is missing an object.

R: Thanks for noticing this error. We have corrected this sentence in the revised manuscript.

l. 164 Please refer to EIS as an indicator of static stability rather than equating both (see above major points).

R: Thank you for this suggestion. We have modified the revised manuscript to refer separately to the concepts of EIS and static stability in the correct sense.

ll. 159-175 Please provide correlation maps (see above major points).

R: We have provided correlations maps (Figure 4) in the revised manuscript according to the point 2 of major issues.

l. 167 Please change "44" to "4".

R: Suggested revision made.

ll. 171-172 Please check for redundancy is both sentences.

R: We have rewritten both sentences to clarify that closed MCC clouds are linked to larger EIS values compared to open MCC clouds, which is consistent with the correlation maps of the revised manuscript.

ll. 181-183 Does this mean the portion of "other" has increased?

R: Taking into account the definition of the category "Other", when the frequency of occurrence of open and closed MCC types reduces during the summer, it leads to a corresponding increase in the proportion of the category "Other" for this season. This relationship can be attributed to the fact that the category "Other" encompasses all coverage types apart from Open and Closed MCCs, causing its relative representation to grow as the frequencies of the specific MCC types decline during the summer months.

ll. 196-198 Please elaborate – currently unclear how land masses explain seasonality.

R: Thanks for noticing this. What we suggested is that the differences for the same season, either winter or summer, between the SO and NP regions can be as well attributed to the contrasting land mass surfaces between both hemispheres. We have rephrased this idea in the revised manuscript.

ll. 196-201 Perhaps a sketch or cartoon image would be useful here to illustrate the point.

R: While we appreciate the suggestion, we believe that a sketch or cartoon image might not be the most effective approach to illustrate the point. Instead, we have incorporated the seasonal variation of the SST gradient and reworded the sentence for clarity. This, combined with the correlation maps, should enhance the clarity of the study. Such analyses provide a more robust and detailed understanding of the spatial patterns and behaviors of the MCC cloud types during both summer and winter seasons.

ll. 200-201 Please rephrase. Currently hard to follow. Substitute "that" with "than".

R: We thank you for this comment. We have rewritten the paragraph to explain the characteristics of open and closed MCC distributions during wintertime.

ll. 199-200 Are lower SST connected to a greater frequency of open MCCs?

R: Based on the correlation maps presented in the revised manuscript, high frequency of open MCC clouds is not directly linked to lower SST values. While higher latitudes show stronger correlation coefficients, the frequency of open MCC clouds is actually lower in these regions. However, an important finding is the strong correlation between the frequency of open MCC clouds and the $M$ index, which includes the difference between SST and air temperature. This observation suggests that the occurrence of open MCC clouds is influenced by the presence of MCAOs. Hence, it is the MCAOs' presence, rather than the absolute SST values, that plays a more significant role in shaping the frequency of open MCC clouds.

Fig. 7 Please explain how and why these regions were selected. Is the diurnal cycle of "closed" similarly and oppositely seen in "other" (given that "open" remains unchanged)?

R: Both regions were selected based on areas with a higher frequency of occurrence for both open and closed MCC clouds compared to surrounding areas. This approach ensured that the daily cycles of both categories were as representative as possible within each region. Regarding the diurnal cycle of the category "Other", as previously mentioned, this category includes all other coverage types, making it challenging to derive a specific diurnal cycle for any particular coverage type. Additionally, JAXA's Himawari-8 cloud products are not available during nighttime, which hinders the separation of low-level clouds within the category "Other" to estimate a daily cycle of these clouds.

Fig. 7 Is there a diurnal cycle in EIS or M?

R: We have added a sentence indicating that the diurnal cycles of the $M$ index and EIS do not display a discernible signal or pattern throughout the day. Furthermore, the Supplementary Material includes the diurnal cycles for both EIS and the $M$ index.

ll. 231-233 Please rephrase, perhaps to "The geolocation of cloud types matches those in other studies".

R: Thank you for this suggestion. We have rewritten the sentence to explain that the distribution of open and closed MCC clouds is consistent with other studies.

l. 233 With respect to "most prevalent", isn't the "other" class even greater in proportion?

R: The category "Other" is the most prevalent as it includes everything else besides open and closed MCC clouds. To avoid confusion, we have rephrased the sentence to indicate that open MCC clouds are more prevalent than closed MCC in both regions.

l. 235 Please check for redundancy (i.e., a similar content in sentence before).

R: Thank you for this suggestion. We have revised the sentence to eliminate redundancy.

l. 239 Perhaps is it worth noting in Section 2 how single and multi-layer clouds are handled in this study.

R: Geostationary satellites such as Himawari-8 cannot see multi-layer clouds in detail due to their viewing angle and the limitations of their remote sensing instruments. Geostationary satellites are positioned at a fixed point in the sky above the equator and continuously observe the same region of the Earth's surface. The viewing angle from the satellite's fixed position does not provide sufficient depth perception to identify individual cloud layers with precision. As a result, geostationary satellites tend to treat multi-layer clouds as a single, integrated cloud mass, leading to limited insights into the cloud structure and properties of each layer. Therefore, we have not handled multi-layer clouds due to the impossibility of Himawari-8 to identify these cloud structures.

l. 241 Unclear whether a specific type is being referred to or not.

R: In the revised manuscript, we have specifically referred in this sentence to the focus clouds of this study, corresponding to open and closed MCC clouds.

l. 243 Please check position of parenthesis here. Perhaps rephrase to "turbulent heat and moisture (i.e., sensible plus latent heat) fluxes".

R: Thank you for this suggestion. We have changed the sentence as suggested by the referee.

l. 244 Perhaps better "frequency of occurrence" instead of "presence".

R: Thank you for this suggestion. We have changed the word "presence" by "frequency of occurrence".

l. 246 Please rephrase sentence for better understanding.

R: We have rewritten the sentence to indicate that the Kuroshio region and its western boundary current are distinguishable for exhibiting notably high frequencies of marine cold air outbreaks.

ll. 258-259 Please elaborate on "showing a strong relationship to the SST gradient". Is SST gradient shown anywhere? How was a "strong relationship" detected?

R: Thank you this comment. In the revised manuscript, we have now included the annual and seasonal SST gradients in Figure S1. The SST gradients presented in Figure S1 demonstrate the variations between winter and summer, revealing a noticeable decrease in SST gradients during the summer months compared to winter. Furthermore, the highest values of the gradients are observed in the annual average and during winter for both regions. These results provide valuable insights into the seasonal changes in SST gradients and highlight the prominent influence of the SO and NP regions on the overall spatial patterns of SST variations.

l. 264 Please elaborate on "well-formed open cells". Would not-so-well-formed open cells be problematic for the classification?

R: We have provided in Section 2 a sentence where we mentioned the criteria to identify open MCCs as explained in Lang et al. (2022). Open MCCs were identified as well-formed rings of low-level clouds arranged into distinctive patterns of hexagonally shaped cells with a clear region in the center.

l. 268 Please quantify "high correlation" (see above major points).

R: Thank you for this comment, as we mentioned it, we have included maps of correlations in the revised manuscript to be consistent with the mention of correlations between MCC clouds and the meteorological indices and variables.

l. 278 Please elaborate why other techniques and other classes are needed.

R: Thank you for your comment. As mentioned earlier, we have included maps of correlations in the revised manuscript to ensure consistency with the discussion of correlations between MCC clouds and the meteorological indices and variables. This addition allows to quantitatively assess the relationships between different variables and MCC cloud types, providing a more comprehensive analysis of their spatial patterns and behavior. We appreciate your valuable input, which has enhanced the clarity and robustness of our research.

ll. 280-281 Please briefly explain whether HIMAWARI has a precipitation product.

R: The Japan Meteorological Agency's Himawari-8 geostationary meteorological satellite does not provide a precipitation product. We have expanded the description of the Himawari-8 satellite in Section 2.1 and added a sentence on the satellite products provided by this.

**References:**

Lang, F., Ackermann, L., Huang, Y., Truong, S. C., Siems, S. T., and Manton, M. J.: A climatology of open and closed mesoscale cellular convection over the Southern Ocean derived from Himawari-8 observations, Atmospheric Chemistry and Physics, 22, 2135–2152, doi:10.5194/acp-22-2135-2022, 2022.

McCoy, I. L., Wood, R., and Fletcher, J. K.: Identifying Meteorological Controls on Open and Closed Mesoscale Cellular Convection Associated with Marine Cold Air Outbreaks, Journal of Geophysical Research: Atmospheres, 122, 11,678–11,702, doi:10.1002/2017JD027031, 2017.

Wang F., Zhang L., Feng J. and Hu D. (2022) Seasonal variability of the North Equatorial Current–Kuroshio Current–Mindanao Current based on observations. *Front. Mar. Sci.* 9:1023020. doi:10.3389/fmars.2022.1023020

Yuan, T., Song, H., Wood, R., Mohrmann, J., Meyer, K., Oreopoulos, L., and Platnick, S.: Applying deep learning to NASA MODIS data to create a community record of marine low-cloud mesoscale morphology, Atmospheric Measurement Techniques, 13, 6989–6997, doi:10.5194/amt-13-6989-2020, 2020.

---

## Author Comment (AC2)

*Referee #2:*

**Summary:**

Lang et al. present an expansion of their earlier study, Lang et al. 2022, where they used a convolutional neural network to identify open and closed mesoscale cellular convective clouds in the Southern Ocean. This study adds clouds in an additional region, the North Pacific, that have been identified with their algorithm (applied to the geostationary Himawari satellite and using brightness temperature, which enables identifications over the full diurnal cycle). Maps of cloud occurrence frequency are contrasted with maps of stability metrics in the two regions annually and seasonally. Visual relationships to regional environmental factors (e.g., Kuroshio current, oceanic polar front, storm track) are documented. The diurnal cycle is also presented for these regions annually and seasonally. Differences in behavior in the North Pacific are qualitatively documented and contrasted with the previously published Southern Ocean behaviors.

**General comments:**

The premise of this study is exciting, and the data developed as part of this and Lang et al. 2022 is quite valuable. It is especially noteworthy and novel to examine the diurnal cycles of MCC cloud types in these two hemispheres. The figures developed in this analysis are very well done, clear and compelling. However, the analysis is limited to qualitatively documenting the behaviors in these regions along with their visual correspondence to stability metrics and sea surface temperature. This provides some insights about regional differences but without quantitative analysis the conclusions are limited, similar to those presented in previous studies, and ultimately not as substantive as they have the potential to be. However, I think the authors can develop this analysis into a valuable contribution to the field and fully realize the potential of this work.

1. My main recommendation is to add quantitative comparisons to bolster the qualitative comparisons and help with interpreting/establishing the differences between the NP and SO regions. "Correlations" are currently discussed but they are based on visual comparisons and not calculated/provided. Actual correlations, between occurrence frequency and meteorological variables, could be calculated at many scales (e.g., within spatial map grid boxes, for annual and seasonal relationships, for composite differences, for diurnal cycles, etc.). By quantifying the relationships that you suggest here, you would greatly strengthen your results and better support your conclusions.

R: We appreciate the valuable feedback and agree with the suggestion. In response, we have conducted a thorough analysis in the revised manuscript, including the calculation of correlation maps (new Figure 4). These correlation maps ensure consistency in evaluating the relationship between both categories of MCC and the meteorological indices $M$ and EIS. Furthermore, we have expanded our investigation to encompass correlation maps for near-surface wind speed and sea surface temperature (SST). These quantitative analyses provide insights into the associations between monthly MCC cloud frequency and monthly averages of $M$ index, EIS, SST, and near-surface wind speed in both study regions. The correlations were computed using the monthly mean MCC cloud frequencies in each grid box and exhibit statistical significance at a 95% confidence level for a 36-point correlation. We have incorporated this quantitative analysis throughout the results and discussion sections, as suggested by the referee. By including these correlation maps, we have gained a deeper understanding of the relationships between the meteorological variables and both open and closed MCC cloud types, thereby enriching our knowledge of their spatial patterns and behavior. The inclusion of these additional analyses enhances the robustness and significance of our findings, and we sincerely thank the referee for contributing to the improvement of our research.

2. The diurnal cycle analysis in this and Lang et al. 2022 is novel and has a lot of potential. However, these results are currently limited to qualitatively documenting the differences between type, season, and region. There is an opportunity here to add more depth to the analysis by quantifying the connections to the

meteorological environment (as you do qualitatively in the first part of the paper). This would lead to a deeper understanding of what is contributing to these diurnal cycle differences through understanding how these cloud types are responding to their environmental diurnal cycles. Being able to interpret why you see differences between regions, seasons, etc. in the MCC diurnal development cycle would be a very valuable contribution to the field.

R: We greatly value this comment and concur with the potential of a study focused on the influences of the meteorological environment on the diurnal cycle of MCC clouds. This would help in understanding what drives the diurnal cycle for each type of MCC and the differences between each region. As mentioned in the manuscript, our first step would be to expand the number of categories to disorganized MCC clouds and no MCC clouds. This would allow us to consider the entirety of low-level clouds as defined in Wood and Hartmann (2006). Once this is achieved, we believe that a detailed analysis of the diurnal cycle for all categories, considering the meteorological factors used in this study, would provide valuable insights into this diurnal cycle analysis.

For instance, we aspire to conduct a study similar to the one by Vial et al. (2021), which provided a detailed analysis of the influence of meteorological variables on the diurnal cycle of mesoscale clouds over the North Atlantic. However, we believe that an essential first step towards this goal is to enhance and refine our Convolutional Neural Network (CNN) by incorporating additional categories of mesoscale cloud organization.

To enhance our analysis of the diurnal cycle, we have proposed several explanations for the distinct diurnal cycles observed in the two regions. However, it is crucial to understand that these factors are intricately interconnected and complex. Therefore, a more comprehensive and detailed investigation is necessary to accurately determine their specific influences.

**Specific Comments:**

Throughout: Please only discuss correlations or variables being "correlated" when you have computed a correlation coefficient and statistically tested whether they are correlated (e.g., p-value ≤ 0.01 for 90% confidence). Visually similar maps and cycle plots are not correlations.

R: We thank you for this comment. We fully agree, and as a result, we have included correlation maps for EIS, $M$ index, SST, and near-surface wind speed in the revised manuscript. Additionally, we have incorporated a detailed discussion of the correlations between these variables and both the open and closed MCC categories. This addition provides a comprehensive analysis of the relationships between the meteorological parameters and both cloud types, enhancing the overall findings and understanding presented in the study.

Line 35: I would suggest removing "are most common in" since sub-tropical decks also have a lot of MCC. Agreed that these MCC types dominate the storm tracks (also see Agee et al. 1973, McCoy et al. 2023 for climatology).

R: Thank you for this suggestion. We have removed the phrase "are most common in" in the revised manuscript. We have also included the references of Agee et al. (1973) and McCoy et al. (2023) in the paragraph. These references provide additional context and support to show that MCC types dominate the storm tracks.

Line 37: Fletcher et al. 2016 is for clouds in general, not MCC. Atkinson and Zhang 1997 and Wood 2012 review this MCC-CAO relationship in detail and McCoy et al. 2017 quantified it more recently.

R: We thank you for this comment. We agree that Fletcher et al. (2016) did not study MCC clouds, and it was included as a reference to marine cold air outbreaks. As suggested, we have modified the references, removing Fletcher et al. (2016), and instead added Atkinson & Zhang (1996) and McCoy et al. (2017), which are more appropriate for the context of the sentences.

Section 2.2: I see the rational of only identifying open and closed MCC and throwing everything else into a catch all since it gives you a high quality MCC dataset. However, I do think it would be valuable to at least subdivide the "other" into low, middle, and high clouds and clear sky so that you have an idea of what is happening with the other low clouds (besides the MCC types) in these regions. From the small MCC absolute frequencies that you are working with, there is clearly a lot of the low-cloud behavior that you are missing throughout your analysis and analyzing that could give you valuable context for whether the MCC behaviors are unique and whether their differences are statistically significant from the base behavior.

R: Thank you for your valuable comment. We recognize the significance of analyzing low-level clouds within the category "Other". As previously mentioned in Lang et al. (2020), we are aware of the limitation in our study for not including other low-level cloud types, such as disorganized MCC and no MCC clouds. To address this limitation and improve our CNN, we plan to incorporate more training samples from both regions to include these additional categories. This approach aims to improve the accuracy and robustness of our CNN model, enabling a more comprehensive analysis of marine atmospheric boundary layer clouds.

Nonetheless, we have addressed this limitation by extending our analysis to include the low-level clouds within the category "Others" and comparing them with the open and closed MCC categories using the Himawari-8 cloud products. The Japan Aerospace Exploration Agency (JAXA) provides Himawari-8 cloud products, such as cloud-top height and cloud optical thickness; however, these data are not available during nighttime. Therefore, to define the low-level clouds in the category "Others", we applied a filter that considers only the diurnal cloud product, identifying clouds below 3.5 km and selecting daytime open and closed MCC. The results of this analysis are presented in Section 3.4, where we examine and compare the characteristics of these low-level clouds with open and closed MCC. This additional analysis allows for a more comprehensive understanding of the cloud patterns in the category "Others" and their relationship with open and closed MCC.

Section 2.3: It might be beneficial to expand beyond stability metrics (EIS and M) and SST. Clouds in this region respond to a variety of factors in opposing ways (e.g., Scott et al. 2020) and MCC are thought to be sensitive to more than just stability and SST in their development (e.g., Eastman et al. 2021, 2022). Temperature advection might be especially useful as it would more accurately characterize the surface forcing contribution in these regions. Expanding your meteorological variable space has the potential to quantify novel relationships between MCC (this is a strong dataset for doing this) and the characteristics of these regional environments and could help to better distinguish your analysis from previous work on MCC behavior in these regions (e.g., Muhlbauer et al. 2014, McCoy et al. 2017, Lang et al. 2022).

R: We acknowledge that other variables may have the potential to show a relationship with open and closed MCC morphologies. In our analysis, we have included near-surface wind speed and calculated correlation maps to explore its influence. The decision to include wind speed was influenced by Lang et al. (2022), who speculated on its relationship with the distribution of MCC clouds. Additionally, a recent study by Eastman et al. (2023) established a connection between near-surface wind speed and cloud morphological transitions. While not directly related to temperature advection, wind speed and temperature advection are closely linked in the atmosphere. Temperature advection involves the horizontal movement of air with different temperatures, driven by variations in atmospheric pressure and wind patterns. Wind speed plays a crucial role in determining the rate of air transport, which in turn affects the strength and extent of temperature advection.

Line 134, 158 and Figure 2: Since you use the oceanic polar front as a reference for discussing cloud behavior, it would be useful to have the corresponding annual/seasonal climatological location of the oceanic polar front plotted on these maps. Otherwise, consider citing literature to support these statements and your conclusions.

R: This is a great suggestion. In Section 2.3, we have included pertinent references regarding the location of the oceanic polar front, which has been linked to strong meridional SST gradients, as discussed in Truong et al. (2020) and Dong et al. (2006).

Line 164: It would be very valuable to spatially correlate these figures, as you imply here, and show the results (e.g., as a map of correlation coefficients with significance indicated). As mentioned above, please do not refer to something as "correlated" unless you are providing a correlation coefficient.

R: Thank you for your valuable comment. As we previously stated, we have incorporated correlation maps (new Figure 4) into the revised manuscript to maintain coherence with the discussion of correlations between MCC clouds and the meteorological indices and variables. This inclusion enables to quantitatively evaluate the relationships between various variables and MCC cloud types, facilitating a more comprehensive analysis of their spatial patterns and behavior. We deeply appreciate your input, which has significantly improved the clarity and robustness of our research.

Figure 4 and 5: It might be clearer to see how the regions differ by looking at a difference plot between the NP and SO cases (since you have the data composited to the same space already). You could also consider looking at how the "other-low cloud" category behaves in this space and how it differs from open and closed MCC (e.g., how anomalous the MCC types are from all low clouds).

R: This is a great suggestion. In response, we have included a figure in Section 3.4 of the revised manuscript to compare both regions and the low-level clouds of the category "Other" with open and closed MCC types during daytime. This additional analysis provides valuable insights into the characteristics and behavior of the low-level clouds within the category "Other", enhancing our understanding of their relationships with open and closed MCC. We are thankful for this comment, as it has improved the comprehensiveness of our research.

Line 175: How is the relationship "better"? Please quantify these statements with correlations or other statistics.

R: Thank you for your comment. We agree that a quantitative analysis is necessary to support statements with correlations or other statistics. We have included correlation coefficient maps and conducted analyses based on these correlations to strengthen the validity of our findings. This addition enhances the rigor and reliability of our research.

Line 188-189: Like with the oceanic polar front, it would be helpful to include a corresponding annual/seasonal climatological storm track to show the relationship with the storm track you suggest here. You could also correlate the location with the occurrence frequency and quantify this. Otherwise, please reference literature supporting this statement about the storm track shift and your conclusions.

R: We thank you for this comment. We agree that understanding the location of the storm tracks is vital to ascertain the peak occurrence of open MCC clouds during summertime. Consequently, we have referenced Shaw et al. (2016), which establishes the location of the storm tracks and explains the reasons for their positioning during summer in both hemispheres.

Figure 6: Worth noting somewhere in the text what Figure 6 is showing and adding to the story (it is only mentioned in passing, not explained).

R: Thanks for noticing this. We have rewritten Section 3.2 to complement the analysis with the seasonal two-dimensional histograms of $M$ versus EIS for open and closed MCC categories that are shown in Figure 6, new Figure 7 of the revised manuscript.

Line 192-193: It would help for these types of comparisons if they were quantified. How do you know this is "more relevant", hard to know that from visually comparing the plots. Consider checking the regressions of frequency on these variables (e.g., in a multiple linear regression that accounts for correlations between predictor variables) and looking at spatial correlation maps.

R: Thank you for your comment. We have taken your suggestion into account and added seasonal correlation maps for $M$ index and EIS as Supplementary Material in the revised manuscript (Figure S2). With the inclusion of these seasonal correlation maps, we can now quantitatively assess the seasonal relationships between both meteorological indices and MCC cloud types, resulting in a more comprehensive analysis of their spatial patterns and behavior.

Line 194 (and throughout): You can say it "corresponds" instead of "correlated", but it would be much better to calculate the coefficients and quantify this (see above comments).

R: We have revised this sentence to reflect the incorporation of new seasonal correlation maps for $M$ index and EIS, which can be found in Figure S2 of the revised manuscript. With these additions, we can now quantitatively assess the correlations between MCC clouds and the EIS during the summer season.

Line 199-200: Why does the cooler SST explain the higher open MCC frequency? Please explain and support statement.

R: Thank you for bringing this to our attention. We have revised the sentence in the manuscript to clarify the association between winter SSTs and open MCC clouds.

Line 200-201: What do you mean here? Please explain and support statement.

R: Thanks for noticing this. In the revised manuscript, we have clarified that during wintertime, closed MCCs are more frequently observed at lower latitudes compared to higher latitudes.

Line 206, 215-216, 217: These are very exciting results. Would you be able to extend your meteorological factor analysis to this diurnal cycle analysis as well (something like Vial et al. 2021)? This could really help you to begin interpreting what factors might be driving these cycles (and why they are different between regions).

R: We agree that a comprehensive analysis of the daily cycle, similar to the one by Vial et al. (2021), would be insightful. The question of what drives the diurnal variability in the frequency of MCC cloud occurrences is indeed intriguing. Specifically, understanding the impact of large-scale environmental conditions on the diurnal cycle of MCC clouds is an interesting direction to explore. However, our current methodology does not readily support such an analysis. A crucial step towards this would be enhancing and refining our CNN by integrating additional categories of mesoscale cloud organization. While we recognize the significance of examining the influence of meteorological variables on the diurnal cycle of mesoscale clouds using Himawari-8, such an investigation necessitates a separate study and falls beyond the scope of the current one.

Line 232-233: Also MODIS (Muhlbauer et al. 2014, McCoy et al. 2017, McCoy et al. 2023).

R: Suggested revision made.

Line 249-251, 263-264, abstract: It seems that you are referring to M as if it is surface forcing or an air-sea temperature difference and contrasting it against static stability as measured by EIS. This is confusing since M is also a stability estimate (essentially a modified form of LTS). This is also inconsistent with your earlier discussions. M can be written as a function of EIS and the air-sea temperature difference (McCoy et al. 2017), is that what you are referencing? Please clarify your meaning and be more accurate in your language.

R: We appreciate your keen observation. In response, we have revised the abstract and pertinent sections of the manuscript to clarify that the *M* index is an effective tool for identifying instances of marine cold air outbreaks. It achieves this by integrating considerations of both surface forcing and lower tropospheric stability.

Line 269: Please explain how you come to this conclusion. Hard to tell from comparing Figure 2 and 5, is this based on a different analysis?

R: Thank you for bringing this to our attention. Based on the seasonal correlation maps between closed MCCs and the EIS, we have rephrased the sentence in the revised manuscript to provide further clarification and support for our discussion on the seasonal differences between open and closed MCCs and their association with the EIS.

Line 272-276: Great to include the discussion on lines 272-274 of why you have these diurnal cycle differences. It would be very valuable to extend this further to the intriguing regional differences you document (i.e. to add interpretation to Lines 274-276).

R: We appreciate your comment. We have provided several interpretations for the observed differences in diurnal cycles between the two regions. However, it is important to note that these elements are interconnected and complex, and a deeper, more detailed investigation is needed to accurately discern their specific influences.

Line 277: How are you quantifying "good performance" here? You previously tested and trained on the SO data (and that is presented well in Lang et al. 2022), are you able to similarly check the accuracy for the NP? Or is this from visual inspection?

R: Thank you for your valuable comment. As highlighted in the first paragraph of Section 3 (Results), the training data for the model was exclusively obtained from the SO. However, following an exhaustive visual inspection of the cloud cover over the NP region, we were able to confirm that the algorithm consistently produced robust and reliable results across both hemispheres. However, we recognize that the phrase "good performance" can be ambiguous and potentially lead to confusion. Therefore, to ensure clarity, we have decided to remove this sentence from the revised manuscript.

Line 280-281: Extending this diurnal analysis, either here or in a future paper, would be fantastic. Your results raise so many interesting questions: why are closed MCC NP cycles smaller than SO? Why are they especially small in the summertime? Why are the closed MCC cycles peaking in magnitude in different seasons in the two regions? Why are open MCC cycles much smaller and peaking at different times? What is happening with the remaining contribution of clouds (your "other" type)? You could make a good start at answering these by quantifying the cycle relationships to the meteorological variables you discussed in the first part of the paper.

R: We deeply appreciate your feedback, particularly concerning the expansion of our diurnal cycle study. We are keen to explore the factors influencing the diurnal variability of MCC cloud occurrences and the contrasts between the NP and SO regions. Alongside the referee's insights, we share an interest in understanding the roles of other low-level clouds, including disorganized MCC and no MCC clouds. As we have highlighted, our next step involves enhancing our CNN to encompass a broader range of mesoscale cloud categories. This enhancement will lay the groundwork for a thorough analysis of all mesoscale cloud dynamics and a more in-depth study of the diurnal cycle.

**Technical Corrections:**

Line 158: "but is relatively"

Line 167: "Figure 4"

Line 185: "considerably"

Line 246: "current in the Kuroshio region"

Line 248: "MCCs than in the SO"

R: All the technical corrections have been made.

**References:**

Agee, E. and Dowell, K.: Observational studies of mesoscale cellular convection, Journal of Applied Meteorology and Climatology, 13, 295 46–53, doi:10.1175/1520-0450(1974)013<0046:OSOMCC>2.0.CO;2, 1974.

Atkinson, B. W. and Zhang, J. W.: Mesoscale shallow convection in the atmosphere, Reviews of Geophysics, 34, 403–431, doi:10.1029/96RG02623, 1996.

Dong, S., Sprintall, J., and Gille, S. T.: Location of the Antarctic polar front from AMSR-E satellite sea surface temperature measurements, Journal of Physical Oceanography, 36, 2075–2089, doi:10.1175/JPO2973.1, 2006.

Eastman, R., McCoy, I. L., & Wood, R.: Wind, rain, and the closed to open cell transition in subtropical marine stratocumulus. Journal of Geophysical Research: Atmospheres, 127, e2022JD036795. doi:10.1029/2022JD036795, 2022

Fletcher, J., Mason, S., and Jakob, C.: The climatology, meteorology, and boundary layer structure of marine cold air outbreaks in both hemispheres, Journal of Climate, 29, 1999–2014, doi:10.1175/JCLI-D-15-0268.1, 2016a.

Lang, F., Ackermann, L., Huang, Y., Truong, S. C., Siems, S. T., and Manton, M. J.: A climatology of open and closed mesoscale cellular convection over the Southern Ocean derived from Himawari-8 observations, Atmospheric Chemistry and Physics, 22, 2135–2152, doi:10.5194/acp-22-2135-2022, 2022.

McCoy, I. L., Wood, R., and Fletcher, J. K.: Identifying Meteorological Controls on Open and Closed Mesoscale Cellular Convection Associated with Marine Cold Air Outbreaks, Journal of Geophysical Research: Atmospheres, 122, 11,678–11,702, doi:10.1002/2017JD027031, 2017.

McCoy, I. L., McCoy, D. T., Wood, R., Zuidema, P., & Bender, F. A.-M.: The role of mesoscale cloud morphology in the shortwave cloud feedback. Geophysical Research Letters, 50, e2022GL101042. doi:10.1029/2022GL101042, 2023

Shaw, T., Baldwin, M., Barnes, E. et al. Storm track processes and the opposing influences of climate change. Nature Geosci 9, 656–664 (2016). doi:10.1038/ngeo2783

Truong, S., Huang, Y., Lang, F., Messmer, M., Simmonds, I., Siems, S., and Manton, M.: A climatology of the marine atmospheric boundary layer over the Southern Ocean from four field campaigns during 2016–2018, Journal of Geophysical Research: Atmospheres, 125, e2020JD033 214, doi:10.1029/2020JD033214, 2020.

Vial, J., Vogel, R., & Schulz, H.: On the daily cycle of mesoscale cloud organization in the winter trades. Quarterly Journal of the Royal Meteorological Society, 147(738), 2850-2873, doi:10.1002/qj.4103, 2021.

Wood, R. and Hartmann, D. L.: Spatial variability of liquid water path in marine low cloud: The importance of mesoscale cellular convection, Journal of Climate, 19, 1748–1764, doi:10.1175/JCLI3702.1, 2006.

---

## Author Response (AR2)

Dr. Graham Feingold,
Editor
Atmospheric Chemistry and Physics
November 15th, 2023

Dear Dr. Feingold:

We sincerely appreciate this new opportunity to revise and improve our manuscript on "On the relationship between mesoscale cellular convection and meteorological forcing: Comparing the Southern Ocean against the North Pacific".

In the revised manuscript, we have made comprehensive modifications to address the questions and suggestions put forth by both referee #1 and you.

We thank both referees for their insightful and useful comments. Below are our point-to-point responses to each of your comments and referee #1's comments.

Sincerely,
Francisco Lang

**Editor**

Some minor comments of my own:

1) The NE Atlantic has been the subject of quite a bit of recent MCAO work (ACTIVATE campaign) and while I don't suggest rigorous comparison, briefly addressing commonalities/differences based on published analyses would be a useful addition.

R: Thank you for your suggestion. We have included additional sentences in the discussion section of the revised manuscript, addressing recent studies conducted over the North Atlantic. Specifically, we have noted the similarities in the transition from closed MCC to open MCC clouds and the influence of MCAOs.

2) cloud radius --> cloud droplet (or drop) radius

R: Suggested revision made.

3) The paper would benefit from a careful read to weed out some typos and grammatical errors.

R: Thank you for your comment. We have meticulously revised the new manuscript to eliminate any typos and grammatical errors. We believe this new manuscript represents an improved version of the previous one.

4) Line 3: abstract: ideal --> idealized

R: Suggested revision made.

4) Line 19: required --> able

R: Suggested revision made.

5) Line 247: LTS --> LST

R: Suggested revision made.

6) Line 266 needs work

R: Thank you. We have rewritten the sentence to enhance clarity and improve understanding.

7) Line 333 needs work

R: Thank you. We have rewritten the sentence to enhance clarity and improve understanding.

8) Lines 360/361: repeats "In addition"

R: Thank you for noticing this. We have rewritten the paragraphs to eliminate any errors.

The authors present an improved version of their paper but need to improve further to make a substantial enough contribution. Please find several major issues below that the author should address before publication.

**Major concerns**

The authors connect meteorological variables to open and closed cellular structures and there is little to no information about their expected connection provided so far. There are several aspects that come to mind that I am missing in the introduction:

(1) What are the expected mechanisms leading to open versus closed cellular organization? And is "closed" considered to be a stage prior to "open"?

R: Thank you for your suggestion. In the Introduction of the revised manuscript, we have explored the expected mechanisms leading to open versus closed MCC organization. Specifically, we discussed the influence of atmospheric and thermodynamic factors on MCC morphology. We have also noted that closed MCCs commonly transition into open MCCs, a phenomenon well-documented in studies such as Eastman et al. (2022) and Yamaguchi et al. (2017).

(2) Are these mechanisms expected to be identical in both subtropical and mid-latitude clouds? For example, Scott et al. (2020) lists subsidence as an important parameter in the subtropics – could that be relevant for the North Pacific and Southern Ocean?

R: Expanding upon the previous suggestion, in the Introduction of the revised manuscript, we have included a comprehensive comparison of open and closed MCC clouds in the midlatitudes, high latitudes, and subtropics. We have highlighted both the differences and similarities in the behavior of these cloud types. For instance, we discussed how the seasonal cycle of closed MCC clouds varies across latitudinal regions. In the midlatitudes, this cycle is influenced by surface forcing, while in the subtropics, it is more closely tied to lower tropospheric stability. The importance of regional environmental factors in shaping MCC cloud characteristics is emphasized by this distinction.

To better embed the results into the state-of-the-art research, the author should then reflect on these expectations in their discussion. For example, do the authors cover all parameters connected to the expectations or are there parameters missing (e.g., parameters indicative of cloud microphysics)? And importantly, are the correlations unambiguously pointing at certain mechanisms or are there several parameters with high correlation (i.e., regions where parameters are well-correlated with one another)?

R: Thank you for you comment. In the revised Introduction, we have explored previous studies that have identified the most important large-scale meteorological and thermodynamic factors that can significantly influence MCC cloud development. Open MCC clouds are particularly sensitive to the most important larger-scale surface forcing, while closed MCC clouds exhibit greater responsiveness to longwave cloud top cooling (McCoy et al., 2017; Wood, 2012). These added references contribute to a comprehensive analysis of the most important larger-scale meteorological parameters driving MCC clouds development. Furthermore, in the Discussion section of the revised manuscript, we have examined additional microphysical parameters that may impact the development of open and closed MCCs.

The authors decided to examine monthly statistics when calculating correlations between meteorological parameter and occurrence frequency, and it is unclear what motivated this decision. What are the expected timescales of parameters driving cloud organization (e.g., are monthly means expected to resolve these rather than hourly data)? Regarding Section 3.3, why look at the diurnal cycle to begin with?

R: The motivation for monthly statistics is because the correlation analysis involves the frequency of occurrence of both types of MCC clouds against the meteorological parameters outlined in the manuscript (EIS, M index, average SST, and near-surface wind speed). Calculating hourly frequency of occurrence is not feasible because hourly data represents Open and Closed MCC as integer numbers, lacking physical meaning, and serving solely to distinguish between cloud types. Furthermore, larger time scales, such as daily frequencies, do not provide a robust sample size for calculating frequencies and correlating them with the meteorological parameters. Thus, we believe that the most robust correlations are derived from monthly frequency of occurrence.

Regarding the diurnal cycle, in Lang et al. (2022), we investigated the diurnal cycle of open and closed MCCs. As stated in the introduction, our goal is to expand this climatology to the NP. We believe that illustrating the differences in the daily cycles between both regions lays the groundwork for future publications to explore these daily cycles in a more detailed and comprehensive manner, as mentioned in the last paragraph of the manuscript as a potential avenue for future research.

The authors sometimes equate correlation with causality (e.g., ll. 184-185, l. 316, and elsewhere). The authors should carefully examine their language throughout the paper.

R: Thank you for your comment. We have revised and rewritten accordingly the new manuscript to avoid any causality regarding the correlations.

The authors included some discussion elements into the results section (e.g., ll. 177-183, and elsewhere). To create a crisper results section that is largely free of speculation, I strongly urge the authors to move discussion material into Section 4.

R: We have exhaustively reviewed and revised the manuscript to ensure the removal of any elements from the results section that belong in the discussion. We believe that the revised manuscript now presents the results and discussion sections more clearly.

**Minor concerns**

l. 5 Replace "roughly" with "evenly" if that is meant here.

R: Suggested revision made.

l. 7 Include "presumed" before "meteorological forcings".

R: Suggested revision made.

13 Please rephrase sentence ("cycle … cycle").

R: Suggested revision made.

l. 27 McCoy et al. (2023) additionally excluded optical depth effects.

R: Suggested revision made.

ll. 70-72 This sentence seems better suited elsewhere.

R: Thank you for the suggestion. We have moved the sentence to Section 2.3, 'Large-scale meteorological indices.

ll. 142-143 Please rephrase "colder SST" with "greater SST", for example.

R: Suggested revision made.

ll. 175-177 These sentences do not fit well here. Please check if ordered (and placed) correctly.

R: Thank you for bringing this to our attention. We have revised and reorganized the paragraphs to improve this section.

l. 136 Please also cite Eastman et al. (2022).

R: In Eastman et al. (2022), MCAOs are not explicitly defined. They did not utilize the $M$ index, making this reference unsuitable for the context of the sentence.

Fig. 4 Please show data points of one selected grid box in a separate scatter plot (perhaps as supporting information) and also plot the interdependence of parameters (windspeed, SST, EIS and M).S

R: We have added scatter plots for selected grid points of every variable in the revised Supplementary Material (Figure S2).

Fig. 1, 3, 4, 5, 6, 7, 8: If "closed" is considered the stage prior to "open" (see major points), better order plots showing "closed" on the left and "open" on the right.

R: Suggested revision made.

Section 1: Perhaps better have one paragraph on identification methods (i.e., uniting ll. 28-33 and ll. 43-49) rather having information scattered throughout.

R: Thank you for your comment. We have slightly reorganized the introduction to consolidate all the information regarding identification methods, which is now presented in the second paragraph.

**Spelling**

l. 54 Replace "region" with "regions".

R: Suggested revision made.

**References:**

Eastman, R., McCoy, I. L., & Wood, R.: Wind, rain, and the closed to open cell transition in subtropical marine stratocumulus. Journal of Geophysical Research: Atmospheres, 127, e2022JD036795. doi:10.1029/2022JD036795, 2022

Lang, F., Ackermann, L., Huang, Y., Truong, S. C., Siems, S. T., and Manton, M. J.: A climatology of open and closed mesoscale cellular convection over the Southern Ocean derived from Himawari-8 observations, Atmospheric Chemistry and Physics, 22, 2135–2152, doi:10.5194/acp-22-2135-2022, 2022.

McCoy, I. L., Wood, R., and Fletcher, J. K.: Identifying Meteorological Controls on Open and Closed Mesoscale Cellular Convection Associated with Marine Cold Air Outbreaks, Journal of Geophysical Research: Atmospheres, 122, 11,678–11,702, doi:10.1002/2017JD027031, 2017.

Wood, R.: Stratocumulus clouds, Monthly Weather Review, 140, 2373–2423, https://doi.org/10.1175/MWR-D-11-00121.1, 2012

Yamaguchi, T., Feingold, G., & Kazil, J.: Stratocumulus to cumulus transition by drizzle: Stratocumulus to cumulus by drizzle. Journal of Advances in Modeling Earth Systems, 9(6), 2333–2349. https://doi.org/10.1002/2017MS001104, 2017

---

## Author Response (AR3)

Dr. Graham Feingold,
Editor
Atmospheric Chemistry and Physics
December 4th, 2023

Dear Dr. Feingold:

We sincerely appreciate this new opportunity to revise and improve our manuscript on "On the relationship between mesoscale cellular convection and meteorological forcing: Comparing the Southern Ocean against the North Pacific".

In the revised manuscript, we have made modifications to address your questions and suggestions.

Below are our point-to-point responses to each of your comments.

Sincerely,
Francisco Lang

**_Editor_**

1) I have some difficulty with the fact that the results are not always clear on differences between how processes play out in the subtropics vs. the mid latitudes in SO and NP. A clear example is that of the SCT, which is a subtropical phenomenon driven by warming SST and may include a role for drizzle (e.g., Yamaguchi, Eastman). The discussion on lines 370-377 mixes subtropical-related work (Eastman) with mid-latitude work (Abel) leading the reader to assume that the mechanisms are the same. Another example of a transition is that of a pocket of open cells which occurs via drizzle, without a clear change in meteorology. A more careful categorization/organization of the role of processes driving transitions is required. I believe this is in line with Reviewer 1's major comments (1) and (2).

R: Thank you for your suggestion. We have revised the paragraph to more clearly distinguish between the processes influencing the subtropics and mid-latitudes. In this revision, we clarify that in the subtropics, the transition from closed to open MCC clouds often occurs under uniform meteorological conditions and is driven by strong winds and intense drizzle, as outlined by Eastman et al. (2022) and Yamaguchi et al. (2017). In contrast, over the mid-latitudes, precipitation significantly influences the transition from closed to open cloud formations during MCAOs, primarily due to the decoupling of the boundary layer, as demonstrated by Abel et al. (2017) and Tornow et al. (2021). We apologize for the lack of clear distinction in the previous version of our manuscript.

2) While I believe there is merit in a monthly data analysis, I am not content with your response to reviewer 1 regarding the choice of monthly rather than daily data. I don't understand the response: "Calculating hourly frequency of occurrence is not feasible because hourly data represents Open and Closed MCC as integer numbers, lacking physical meaning, and serving solely to distinguish between cloud types." If the 'integer' indicates that an open or closed MCC occurred, why would it not be useful for calculating a freq of occurrence? I also don't follow the logic in the next sentence:

"Furthermore, larger timescales, such as daily frequencies, do not provide a robust sample size for calculating frequencies and correlating them with the meteorological parameters. Thus, we believe that the most robust correlations are derived from monthly frequency of occurrence."

First, daily frequencies are shorter timescales, and these would increase the sample size, though probably also increase the 'noise'. Please provide a clearer explanation of your choice of monthly cycles in the text.

R: Our apologies for the lack of clarity in our previous response. In our initial approach to calculating correlations, we used monthly averages, based on the assumption that longer time scales would effectively filter out noise, as suggested by the editor. Furthermore, we aimed to maintain consistency with prior studies, such as those by McCoy et al. (2017) and Muhlbauer et al. (2017), which also analyzed monthly correlations between the meteorological variables used in this study and the frequency of MCC clouds.

Nonetheless, we have calculated the correlations using daily frequencies of MCC clouds and daily averages of the _M_ index, EIS, SST, and near-surface wind speed. The results showed that, while daily correlations exhibit more noise compared to those on a monthly scale, resulting in lower correlations, their spatial patterns still align with the monthly correlations. We have included the daily correlations in the Supplementary Material (Figure S3).

**Other comments:**

Did you intend to remove the sentence prior to the last? The information appears to be the same.

R: Our apologies for the confusion. We rephrased the sentence in the previous revision as requested by referee #1.

Line 48: Somewhere here please also mention POCs — also a transition of closed to open, with no distinct change in meteorology.

R: Thank you for your suggestion. In the introduction, we have noted that in the subtropics, the lack of substantial meteorological differences between open and closed MCC clouds suggests a predominant role for the precipitation mechanism.

Line 49: please clarify that the transition via drizzle discussed by Yamaguchi occurs during advection over warmer ocean temperatures.

R: We have revised the sentences to indicate that one mechanism driving the transition is advection over warmer waters, where drizzle leads to the breakup of closed MCCs into open MCC clouds, as described by Yamaguchi et al. (2017).

Missing important references to ACTIVATE (NW Atlantic) results (e.g. work of Tornow).

R: We have included a sentence about the work of Tornow et al. (2021), which uses a case of MCAO within the Aerosol Cloud meTeorology Interactions oVer the western ATlantic Experiment (ACTIVATE) campaign.

Please clarify what the correlation is between.

R: We have rephrase the sentence to clarify that the correlations is between closed MCC and EIS.

**References:**

Eastman, R., McCoy, I. L., & Wood, R.: Wind, rain, and the closed to open cell transition in subtropical marine stratocumulus. Journal of Geophysical Research: Atmospheres, 127, e2022JD036795. doi:10.1029/2022JD036795, 2022

McCoy, I. L., Wood, R., and Fletcher, J. K.: Identifying Meteorological Controls on Open and Closed Mesoscale Cellular Convection Associated with Marine Cold Air Outbreaks, Journal of Geophysical Research: Atmospheres, 122, 11,678–11,702, doi:10.1002/2017JD027031, 2017.

Muhlbauer, A., McCoy, I. L., and Wood, R.: Climatology of stratocumulus cloud morphologies: microphysical properties and radiative effects, Atmospheric Chemistry and Physics, 14, 6695–6716, https://doi.org/10.5194/acpd-14-6981-2014, 2014.

Yamaguchi, T., Feingold, G., & Kazil, J.: Stratocumulus to cumulus transition by drizzle: Stratocumulus to cumulus by drizzle. Journal of Advances in Modeling Earth Systems, 9(6), 2333–2349. https://doi.org/10.1002/2017MS001104, 2017

Tornow, F., Ackerman, A. S., and Fridlind, A. M.: Preconditioning of overcast-to-broken cloud transitions by riming in marine cold air outbreaks, Atmospheric Chemistry and Physics, 21, 12 049–12 067, https://doi.org/10.5194/acp-21-12049-2021, 2021.

---

## Author Response (AR4)

Dr. Graham Feingold,
Editor
Atmospheric Chemistry and Physics
December 23rd, 2023

Dear Dr. Feingold:

We sincerely appreciate the recommendation to publish our manuscript. In response to the latest feedback, we have made modifications in the revised manuscript to address both comments.

1. We have rewritten the sentence in the introduction to clearly differentiate between pockets of open cells and stratocumulus-to-cumulus transitions in the subtropics. The revised sentence is now as follows:

   *However, this distinction becomes less apparent in subtropical regions. Here, the formation of pockets of open cells in the absence of significant meteorological changes suggests that the precipitation mechanism plays a predominant role (Savic-Jovcic & Stevens, 2008). Additionally, in the subtropics, the transition from stratocumulus to cumulus is driven by warming SST and weakening subsidence, where drizzle also plays a role. This finding indicates a potential zonal variation in the importance of these influencing factors. Closed MCCs commonly transition into open MCCs and disorganized MCCs in the subtropics. One mechanism driving this transition is advection over warmer water, where drizzle leads to the breakup of closed MCCs into open MCC clouds (Eastman et al., 2022; Yamaguchi et al., 2017).*

   Furthermore, as you recommended, we have incorporated the reference to Savic-Jovcic & Stevens (2008) in relation to pockets of open cells in the subtropics. Regarding the discussion, we have rewritten the sentences as follows:

   *In the subtropics, the transition from closed to open MCC clouds is often preceded by strong winds that increase moisture, which leads to more intense drizzle and facilitates the transition (Eastman et al., 2022).*

2. We have rewritten the sentence as you recommended

Sincerely,
Francisco Lang